# Chemical and structural heterogeneity of olive leaves and their trichomes
Victoria Fernández [1,2] ✉, Lisa Almonte[3,4], Héctor Alejandro Bahamonde [5], Ana Galindo-Bernabeu [3,6], Giovanni Sáenz-Arce [3,7] & Jaime Colchero[3]

Many biological surfaces have hairs, known as trichomes in plants. Here, the wettability and macro- and micro-scale features of olive leaves are analyzed. The upper leaf side has few trichomes, while the lower side has a high trichome density. By combining different techniques including electron and atomic force microscopy, trichome surfaces are found to be chemically (hydrophilic-hydrophobic) heterogeneous at the nano-scale. Both olive leaf surfaces are wettable by water, having a high water contact angle hysteresis and great drop adhesion. The ultra-structural pattern observed for epidermal pavement cells differs from the reticulate cuticle structure of trichomes which shows that leaf surface areas may be substantially different despite being located nearby. Our study provides evidence for the nano-scale chemical heterogeneity of a trichome which may influence the functional properties of biological surfaces, such as water and solute permeability or water capture as discussed here for plants.

Plant surfaces provide protection against biotic and abiotic stress factors, such as water loss[1], excess UV radiation[2], pest and pathogen attack[3] or ice formation[4]. Aerial plant organs have an epidermis as outermost protecting tissue which may include modified cell structures like stomata and trichomes, in addition to pavement cells[5,6]. The external cell wall of epidermal cells has a lipid-rich area named cuticle[7,8], which is a composite material made of hydrophobic (lipids; cutin polymer and waxes) and hydrophilic (cell wall carbohydrates) compounds[9]. In addition to cell wall polysaccharides (i.e., cellulose, pectin and hemicellulose polymers), this part of the epidermal cell wall contains cutin polyester and waxes embedded (intra-cuticular) or deposited (epi-cuticular) onto the surface[9,10]. Cutin is mainly composed of C16 and C18 fatty acids, with one or more hydroxyl, mid-chain epoxide and end-chain carboxyl functional groups[11,12]. Cuticular waxes include a mixture of linear, long-chain aliphatic molecules (e.g. alkanes, aldehydes, fatty acids or alcohols) and in some species cyclic compounds such as triterpenoids and aromatics[13,14].

Some studies carried out with few plant species and organs during the last decade, showed the chemical heterogeneity of the cuticle and importance of cell wall polysaccharides[9,10,12,15–18]. The occurrence of hydrophilic nano-areas in the surface of a plant organ

has been recently demonstrated with rose petals as model plant material[19].

The wetting properties of plant surfaces are of interest for the development of bioinspired materials suitable for various purposes[20–25], and may play a major functional role such as water capture, foliar uptake or water delivery to the root[20,26,27]. Leaf surface traits may vary upon factors such as organ side, state of development or growing conditions, and may be rather of totally glabrous (i.e., with no trichomes) or have significant trichome densities[28–30]. Leaf wettability is affected by chemical composition and topography, but currently there is limited information on the nano-scale distribution of cuticle hydrophilic and hydrophobic components[19] which may have a major impact on wetting and potential adherence or repulsion of water deposited onto the foliage as for example, rain, dew or fog[30,31]. Indeed, the importance of leaf surface wettability as an adaptation for enhancing gas exchange, limiting transpiration and improving water use efficiency has received limited scientific attention to date[28,29,32,33]. Adhesion (drops or films) or repulsion of water considering also potential condensation and water capturing mechanisms, may facilitate or prevent the retention and potential transport of water and gases across leaf surfaces which may be a key eco-physiological trait[30,31,34].

[1]Department of Systems and Natural Resources, School of Forest Engineering, Universidad Politécnica de Madrid, C/ José Antonio Nováis, 10, 28040 Madrid, Spain. [2]Centro para la Conservación de la Biodiversidad y el Desarrollo Sostenible, E.T.S.I. Montes, Forestal y del Medio Natural, Universidad Politécnica de Madrid, 28040 Madrid, Spain. [3]Centro de Investigación en Óptica y Nanofísica, Departamento de Física, Campus Espinardo, Universidad de Murcia, 30100 Murcia, Spain. [4]Applied Physics Department, Universidad de Alicante, 03080 Alicante, Spain. [5]Facultad de Ciencias Agrarias y Forestales, Universidad Nacional de La Plata, Diagonal 113 No 469, 1900 La Plata, Argentina. [6]Universidad Técnica Nacional (UTN), Alajuela, Costa Rica. [7]Departamento de Física, Facultad de Ciencias Exactas y Naturales, Universidad Nacional, Heredia 86-3000, Costa Rica. ✉e-mail: v.fernandez@upm.es

Leaf trichomes may provide protection against multiple biotic and biotic stress factors such as excess water loss[35]. They are often found in leaves of plants grown in arid environments and may play a role in dew harvesting and other wetting phenomena[36–38]. The protecting role of the non-glandular, multicellular, peltate trichomes occurring onto the lower side of olive leaves has been analyzed in some investigations[2,39–44]. Peltate trichomes in olive leaves accumulate ultraviolet-B (UV-B, 280-320 nm) absorbing compounds which protect the underlying tissue against damaging UV-B radiation[2,40,41,45].

In theory, a trichome indumentum may increase the thickness of the leaf boundary layer and reduce conductivity to water vapor diffusion[36]. It may also increase reflectance and reduce leaf temperature, lowering leaf transpiration rates[43,46]. However, no evidence for a protecting role of the abaxial olive leaf indumentum regarding water loss has been reported, despite the potential positive effect of trichomes on water use efficiency[36]. Recently, Bei et al.[31] proposed a mechanism by which umbrella-shaped ratchet trichomes of abaxial *Elaeagnus angustifolia* leaf surfaces collect water, but did not consider the possible contribution of surface chemical heterogeneity.

In this study, we aimed at characterizing the wettability and surface characteristics of the upper (adaxial) and lower (abaxial) leaf surface of olive leaves as model plant material, with focus on trichomes. The following hypotheses were tested by measuring leaves of similar age: (i) adaxial and abaxial surfaces have a different degree of wettability and physico-chemical properties owing to the contribution of trichomes, and (ii) olive leaf surfaces are chemically, in addition to structurally heterogeneous.

## Results
### Olive leaf wettability

For assessing the physico-chemical properties of olive leaves, dynamic (only with water) and static (or pseudo-equilibrium) contact angles with liquids of different nature were measured (Fig. 1, Table 1).

Scattered, multicellular peltate trichomes can be observed on the upper (adaxial) leaf side at a density of $21 \pm 2$ per mm$^{-2}$ in younger leaves (Fig. S2A, B). As leaves become older, some of the adaxial trichomes are shed or become buried in the epicuticular waxes (Figs. 1A and S2 C, D).

Trichomes had a mean diameter length of $157 \pm 25$ μm and no side differences were estimated between adaxial and abaxial trichomes. Wettability and microscopic observations were always performed with approximately 6-months-old leaves which had a $21 \pm 2$ density of trichomes per mm$^{-2}$ in the upper leaf side. By contrast, the abaxial side is covered with up to 3 layers of trichomes, with a frequency of circa $106 \pm 9$ trichomes mm$^{-2}$ (Fig. 1D). For the 6-month-old leaves analyzed we estimated a $9 \pm 5\%$

trichome coverage in the adaxial leave side, versus the 100% trichome coverage of the abaxial leaf side (Fig. 1A, D).

The adaxial and abaxial side of olive leaves are wettable for all test liquids with few exceptions, the lowest pseudo-equilibrium contact angles being measured for apolar diiodomethane drops. Advancing contact angles for water were significantly lower compared to the upper leaf side (85 versus 96°), but receding angles were low and within the same range for both leaf sides (Table 1). Both surfaces have a very high water contact angle hysteresis, indicating a great degree of adherence/retention of water to the adaxial and abaxial surface of olive leaves. The major change observed in relation with leaf age is that abaxial side becomes more wettable by water as compared to younger leaves (Table S1).

Assuming that surface macro-roughness is sufficiently low (i.e., the surface is essentially flat and does not modify the contact angle), the dispersive and non-dispersive components of the solid can be estimated from measuring (pseudo-equilibrium) contact angles with different liquids[47]. The surface free energy and related parameters of olive leaf surfaces were calculated[48] as shown in Table 2. Both leaf sides had a total surface free energy ($\gamma_s$) between 30 and 37 mJ m$^{-2}$, with a major contribution of the dispersive/ Lifshitz -van der Waals/apolar component ($\gamma^{LW}$) and a limited contribution of the non-dispersive component which includes polar interactions ($\gamma^{AB}$), chiefly in the case of the upper leaf side. The lower leaf side (fully covered with peltate trichomes) has a higher polarity than the adaxial

**Table 1 | Pseudo-equilibrium contact angles ($\vartheta_0$) of drops of water, glycerol and diiodomethane (DM) with the adaxial and abaxial side of olive leaves**

| Liquid | Surface | Contact angles (°) | | | |
|---|---|---|---|---|---|
| | Leaf side | Equilibrium ($\vartheta_0$) | Advancing ($\vartheta_{adv}$) | Receding ($\vartheta_{rec}$) | Hysteresis ($\Delta\vartheta_{hys}$) |
| Water | Adaxial | 70 ± 5 a | 85 ± 7 a | 30 ± 4 a | 55 ± 6 |
| | Abaxial | 90 ± 6 b | 96 ± 7 b | 29 ± 4 a | 67 ± 6 |
| Glycerol | Adaxial | 72 ± 7 a | – | – | – |
| | Abaxial | 97 ± 5 b | – | – | – |
| DM | Adaxial | 61 ± 6 a | – | – | – |
| | Abaxial | 62 ± 5 a | – | – | – |

For water, advancing ($\vartheta_{adv}$), receding ($\vartheta_{rec}$) contact angles and contact angle hysteresis ($\Delta\vartheta_{hys}$) values are provided. Data are means ± standard deviations (SD).

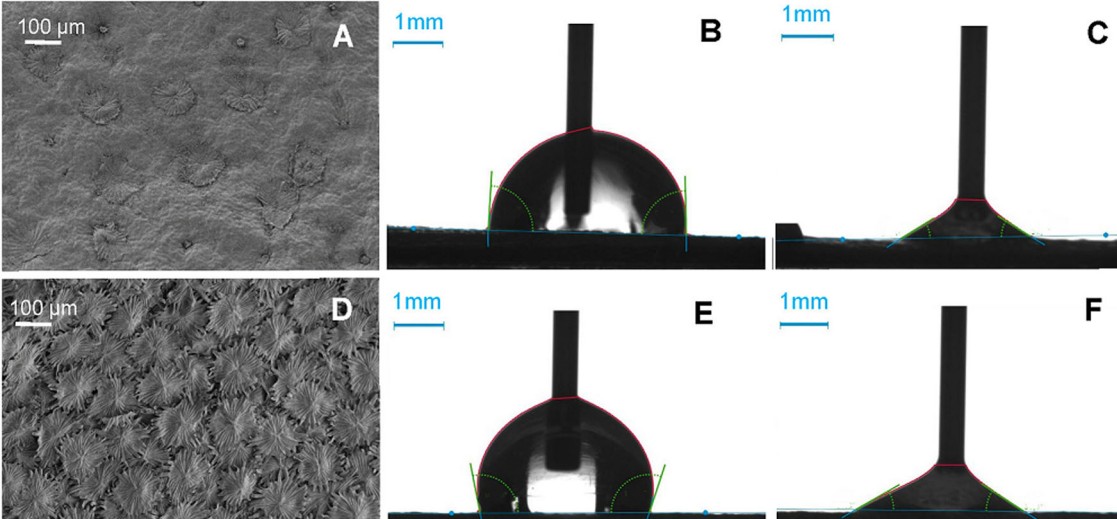

**Fig. 1 | Wettability and topography of olive leaf surfaces.** Adaxial (**A**) and abaxial (**D**) olive leaf surfaces observed by SEM, with examples of advancing ($\vartheta_{adv}$, **B**, **E**) and receding ($\vartheta_{rec}$, **C**, **F**) contact angles of drops water.

**Table 2 | Total surface free energy (γ).** Lifshitz van der Waals component ($\gamma^{LW}$), Acid-base component ($\gamma^{AB}$) with the contribution of electron donor ($\gamma^-$) and electron acceptor ($\gamma^+$) interactions, total surface free energy ($\gamma_s$) and polarity ($\gamma^{AB} \gamma^{-1}$ x 100) of adaxial and abaxial olive leaf surfaces

| Leaf side | $\gamma^{LW}$ (mJ m$^{-2}$) | $\gamma^-$ (mJ m$^{-2}$) | $\gamma^+$ (mJ m$^{-2}$) | $\gamma^{AB}$ (mJ m$^{-2}$) | $\gamma_s$ (mJ m$^{-2}$) | *Polarity* (%) | $\delta$ (MJ$^{1/2}$ m$^{-3/2}$) |
|---|---|---|---|---|---|---|---|
| Adaxial | 28 | 20.26 | 0.08 | 3 | 31 | 8 | 16 |
| Abaxial | 27 | 13.24 | 1.62 | 9 | 37 | 25 | 19 |

**Fig. 2 | Olive leaf cross-sections observed by OM (A–C) and TEM (D, E) after tissue staining.** **A** General leaf structure with focus on the adaxial and abaxial epidermis where trichomes can be observed. **B** Adaxial epidermis with a trichome. **C** Abaxial epidermis with trichomes. **D** Adaxial epidermal pavement cells having a thick cuticle and a layer of epicuticular waxes. **E** Abaxial epidermis showing a longitudinal section of a trichome and a sinuous pavement cell epidermal surface. Letters indicate: UE upper (adaxial) epidermis, LE lower (abaxial) epidermis, UT adaxial leaf surface trichome, LT abaxial leaf surface, CW cell wall, EC epidermal cell, AdC adaxial leaf surface cuticle, AbC abaxial leaf surface cuticle, TC trichome cuticle, TCW trichome cell wall.

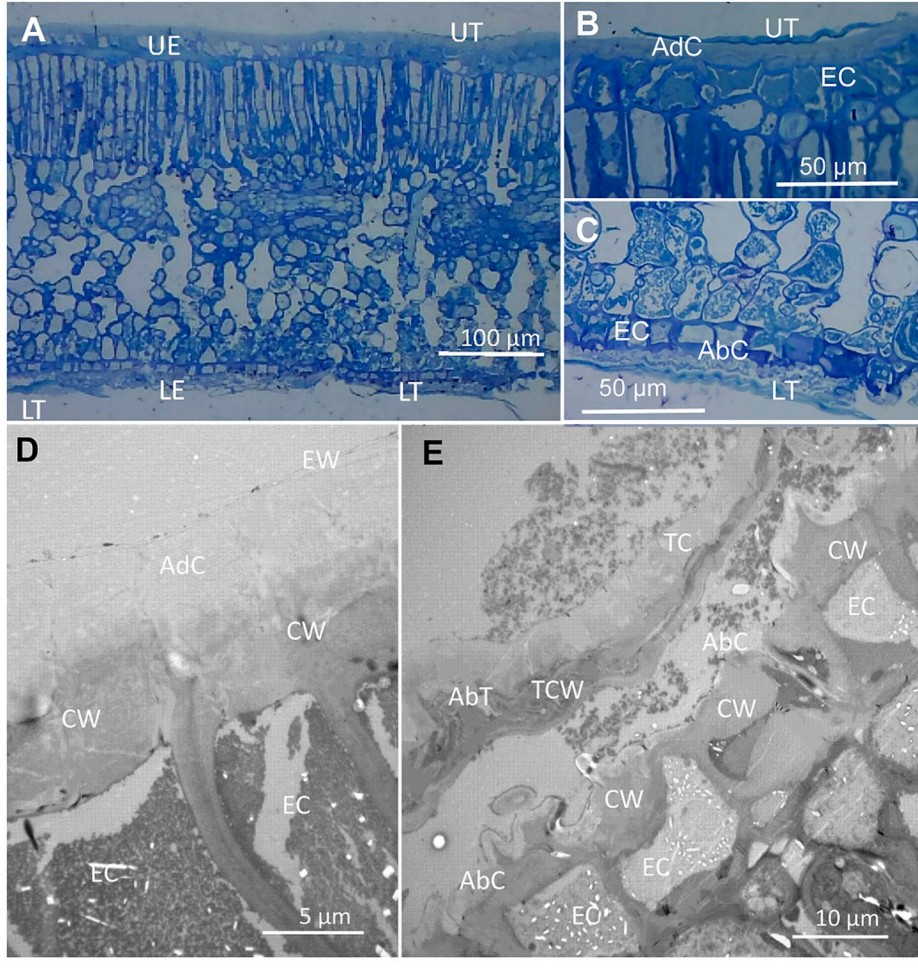

one which is principally covered with epicuticular waxes (Fig. 1A). The solubility parameter (δ) of the abaxial side is higher than that of the upper leaf side. This is also related to the occurrence of peltate trichomes in the surfaces which provide roughness and chemical composition differences between both leaf sides, as described in the following sections. Aging principally involves changes in the abaxial leaf side which has higher $\gamma_s$, polarity and δ values likely associated with trichome surface degradation and exposure of cell wall polysaccharides (Table S2).

## Olive leaf surface structure and chemical heterogeneity

For gaining insight into the cross-section structure of olive leaf surfaces, we carried out transmission electron microscopy (TEM) and optical microscopy (OM) observations, after tissue staining (Figs. 2 and 3). Optical microscopy leaf cross-sections were stained with Toluidine Blue which stains cell wall polysaccharides in blue color[49]. The adaxial side has scattered peltate trichomes (Fig. 2A) that can be seen as blue-stained, elongated structures occurring also in the abaxial surface, but a higher frequency (Fig. 2A–C). In TEM cross-sections which were stained with metals (Figs. 2 D, E and 3), the adaxial cuticle is observed as a comparatively smooth and

thick, electron-lucent area often having a transparent layer of amorphous epicuticular waxes on top and as interface with the surrounding atmosphere (Fig. 2D). The adaxial and abaxial epidermal pavement cell cuticle is mainly amorphous and can be assigned to Type 6 of Holloway´s classification[7], with a layer of electron-lucent epicuticular waxes on top of it (Fig. 2D, E). By contrast, the lower epidermis has a rough and thinner cuticle having cuticular folds (Fig. 2E), in addition to stomata and trichomes as surface features.

Adaxial and abaxial trichomes have a cuticle of variable thickness and ultra-structure along different trichome cross-section areas (Fig. 3). In general, the trichome cuticle has a Type 4 ultra-structure in which all regions are reticulate[7,50]. The cuticle and cell wall at the central trichome area near the base is thicker (Fig. 3A), and gradually becomes thinner at increased distance from the trichome center (Fig. 3B). Both the upper and lower side of the peltate trichome has a cuticle, underlying cell wall material being observed as a dark, electron-dense zone (Fig. 3D). In some areas, the cell/lumen is very thin (see Fig. 3D, E) and the upper and lower cell wall of the trichomes are almost in contact with each other, as observed at the end of the trichome (Fig. 3E). The trichome cuticle has a major degree of reticulation in

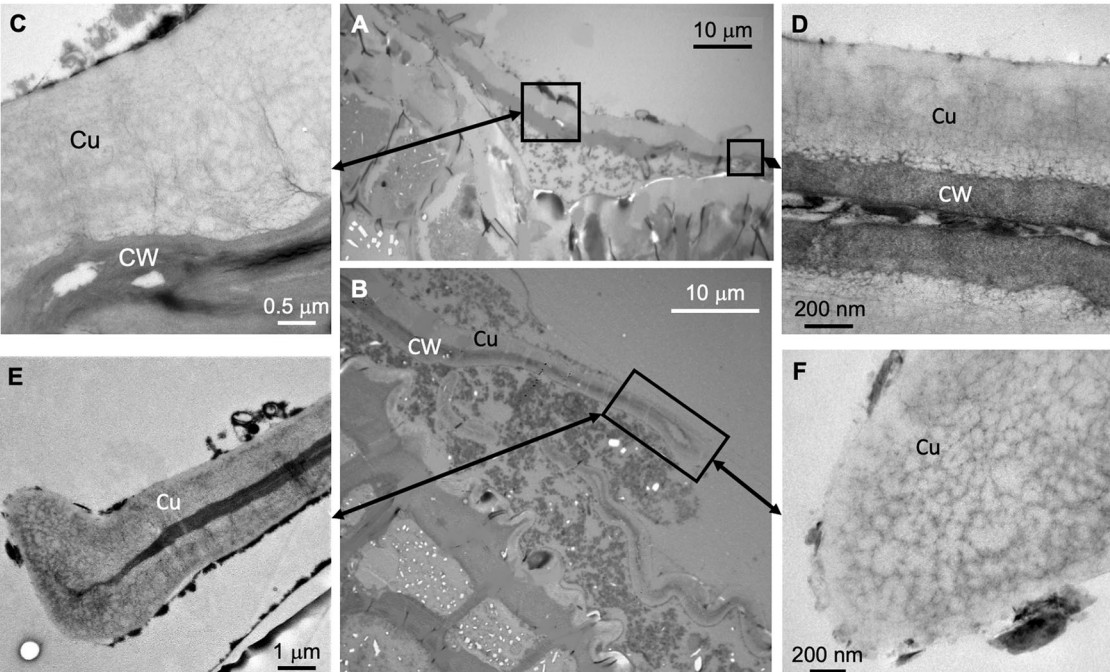

**Fig. 3 | Olive leaf trichome longitudinal cross-sections observed by TEM. A** Base (center) of the trichome where it is anchored to the epidermis. **B** End of a trichome with abaxial epidermal pavement cells underneath. **C** cuticle and cell wall of an area close to the trichome center. **D** Adaxial and abaxial cuticle and cell wall of a trichome located ~30 μm away from the trichome center. **E** End of a trichome. **F** detail of the cuticle of this terminal trichome zone. Letters indicate: Cu cuticle, CW cell wall.

some zones, with the occurrence of dark strands likely associated with polysaccharides at high densities as observed in the trichome edge shown in Fig. 3E, F.

### Trichome AFM characterization

Atomic force Microscopy (AFM) was employed to directly measure the topography $z(x,y)$ of leaf surfaces, simultaneously providing nanoscale maps $ch(x,y)$ related to the chemistry of the sample by analyzing tip-sample interactions[51]. Here, we associated chemical information with the interaction generated by the formation of liquid necks[52,53]. Chemical maps thus describe local hydrophilicity/hydrophobicity. Topography and chemical AFM data were measured on upper and lower olive leaf trichomes, using optical microscopy to precisely align the AFM tip measuring morphology and frequency shift data (Fig. S1).

Figure 4 shows pairs of topography and frequency shift images acquired at different resolutions on an adaxial leaf surface trichrome. The lowest resolution images (Fig. 4A, D) show parts of one trichrome cell which extends radially from the center of the trichome, whose highest central part is about 1 μm higher than the edge of the cell. The width of this cell is about 5 μm and the length of the section observed is about 15 μm, the whole cell being however significantly larger (Fig. S1). At higher resolution (Fig. 4B, E, C, F) we find no clear well-defined morphological features, surfaces being relatively rough The topographic images A, B, and C have a roughness of 260 nm$_{rms}$ (on a 25 μm scale), 60 nm$_{rms}$ (on a 10 μm scale), and 16 nm$_{rm}$ (on a 1 μm scale), respectively.

In comparison to the topographic images of the adaxial side (Figs. S1 and 4A–C), frequency shift images (Fig. 4D–F) show a more detailed structure. The color scale of these images has been chosen so that blue represents a stronger interaction due to attractive forces induced by the liquid necks formed between tip and sample during measurement[52]. Blue color therefore corresponds to more hydrophilic regions, while yellow-red color relates to hydrophobic regions. No correspondence is found between morphology and nanoscale wetting properties, i.e., higher or lower topographic areas may be hydrophilic or hydrophobic. In addition, while the smallest topographic features observed are in the 100–200 nm range (lateral size), the smallest variation of wetting properties occurs on a 25 nm scale,

which is about the maximum resolution we expect for DAFM in this kind of images. Therefore, high and low topographic regions of the surface may be hydrophilic and hydrophobic, and within the same morphological structure we also find hydrophilic and hydrophobic domains.

Concerning the trichomes of the abaxial leaf side, we essentially found that their morphology and nanoscale wetting properties were similar to the features of the adaxial ones (Fig. 5), i.e.: we observed fine details in the frequency shift images and thus a high spatial variation of wetting properties. Again no correspondence between topographic structures and their nanoscale wetting properties was found, as described above. In this context, we note that the large-scale image Fig. 5A (topography) and the corresponding image D (wetting) seem to indicate that higher regions are more wettable (bluer in the central region of image D). We however remain cautious, and do not want to over-emphasize the significance of these data, since large-scale AFM images are more difficult to be obtained and correctly interpreted due to AFM-related issues (such as e.g., non-ideal feedback, piezo hysteresis and large topographic corrugation; among others).

To compare the morphology and the wetting properties of the background material of the upper leaf surface (i.e., epicuticular waxes), AFM images were acquired near the edge of a trichrome, as shown in Fig. 6. To avoid the measuring drawbacks discussed above for large-scale AFM images, a 50 μm × 10 μm stripe of the adaxial leaf side was imaged close to the border of a trichrome, simultaneously showing the trichome edge and the lower, epicuticular waxes covering epidermal pavement cells. Note that since trichomes completely cover the abaxial olive leaf side with the occurrence of various trichome layers (Fig. 1D), images simultaneously showing a trichrome and epicuticular waxes could not be acquired by AFM, because only trichomes were measured (see, cross-sections in Figs. 2 and 3).

From the data shown in Fig. 6, we conclude that the upper leaf side background material (i.e., epicuticular waxes) has a lower roughness than the trichome surface, and is – on average – less wettable than the trichomes. In addition, the variability of the wetting properties of this background waxy region is smaller than the variations on the trichrome, i.e., the mean value of wetting as well as the "roughness" of the wetting images (Fig. 6D–F) is larger on the trichrome as compared to the surface epidermal pavement cells.

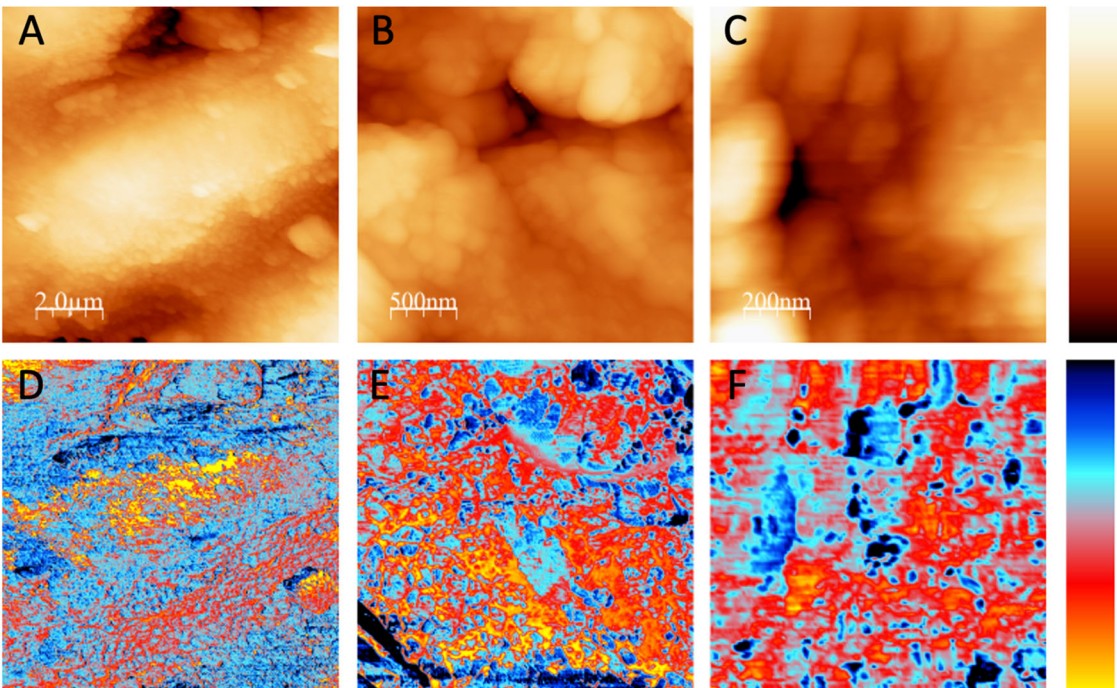

**Fig. 4 | AFM images of regions of an adaxial olive leaf trichrome at different lateral resolutions.** The upper row shows topographic images (**A–C**) and the lower row corresponds to frequency shift data (**D–F**). Left images (**A, D**): 10 μm × 10 μm lateral size, the total color scale is 1.5 μm for topography and 400 Hz for frequency shift. Central images (**B, E**): 2.5 μm × 2.5 μm lateral size, 400 nm topography and 400 Hz frequency shift. Left images (**C, F**): 1 μm × 1 μm lateral size, 80 nm and 160 Hz frequency shift.

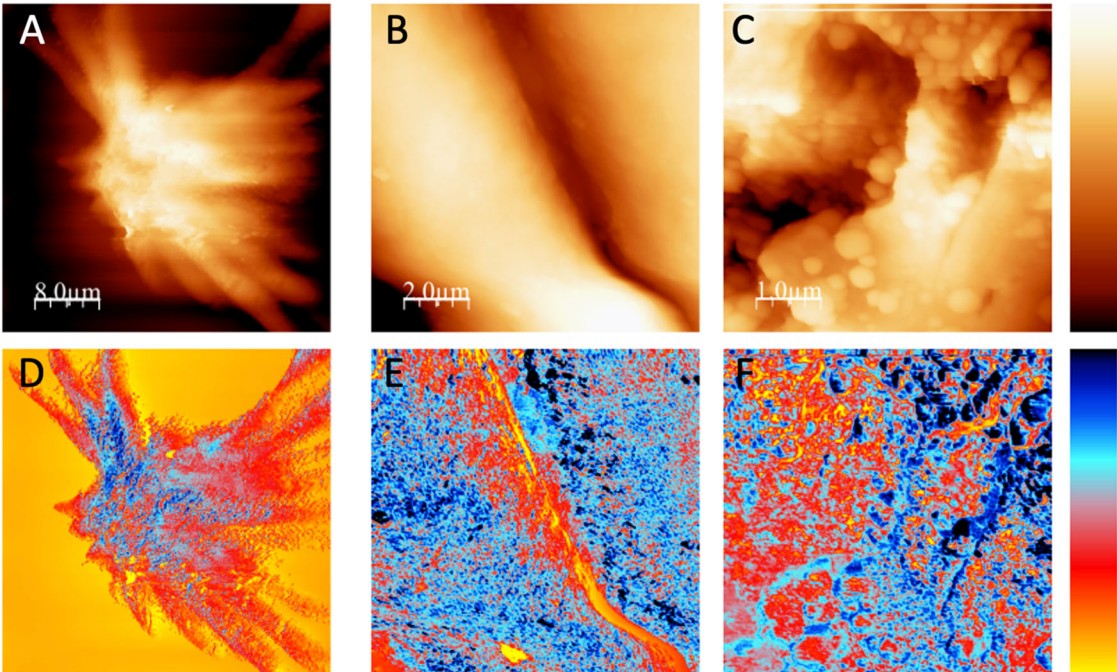

**Fig. 5 | AFM images of regions of an abaxial olive leaf trichrome at different lateral resolutions.** The upper row (**A–C**) shows topographic images and the lower row (**D–F**) corresponds to frequency shift data. Left images (**A, D**): 40 μm × 40 μm lateral size, the total color scale is 5 μm for topography and 400 Hz for frequency shift. Central images (**B, E**): 10 μm × 10 μm lateral size, 2.5 μm topography and 200 Hz frequency shift. Left images (**C, F**): 5 μm × 5 μm lateral size, 600 nm topography and 200 Hz frequency shift.

## Trichome surface chemical heterogeneity

For evaluating the potential chemical heterogeneity of the adaxial and abaxial olive leaf surfaces, approximately 3-months-old olive leaves (still having a high trichome density in the upper side as observed in Fig. 7A) were immersed in solutions containing cellulose and pectin degrading enzymes (i.e., cellulase and pectinase, respectively) compared to leaves kept in water (control treatment) for 8 days and then observed potential surface changes by SEM. We noticed that olive leaf surfaces were attacked by cellulase and pectinase enzymes and observed that some upper side trichomes were shed (see arrows in Fig. 7C). While it is not

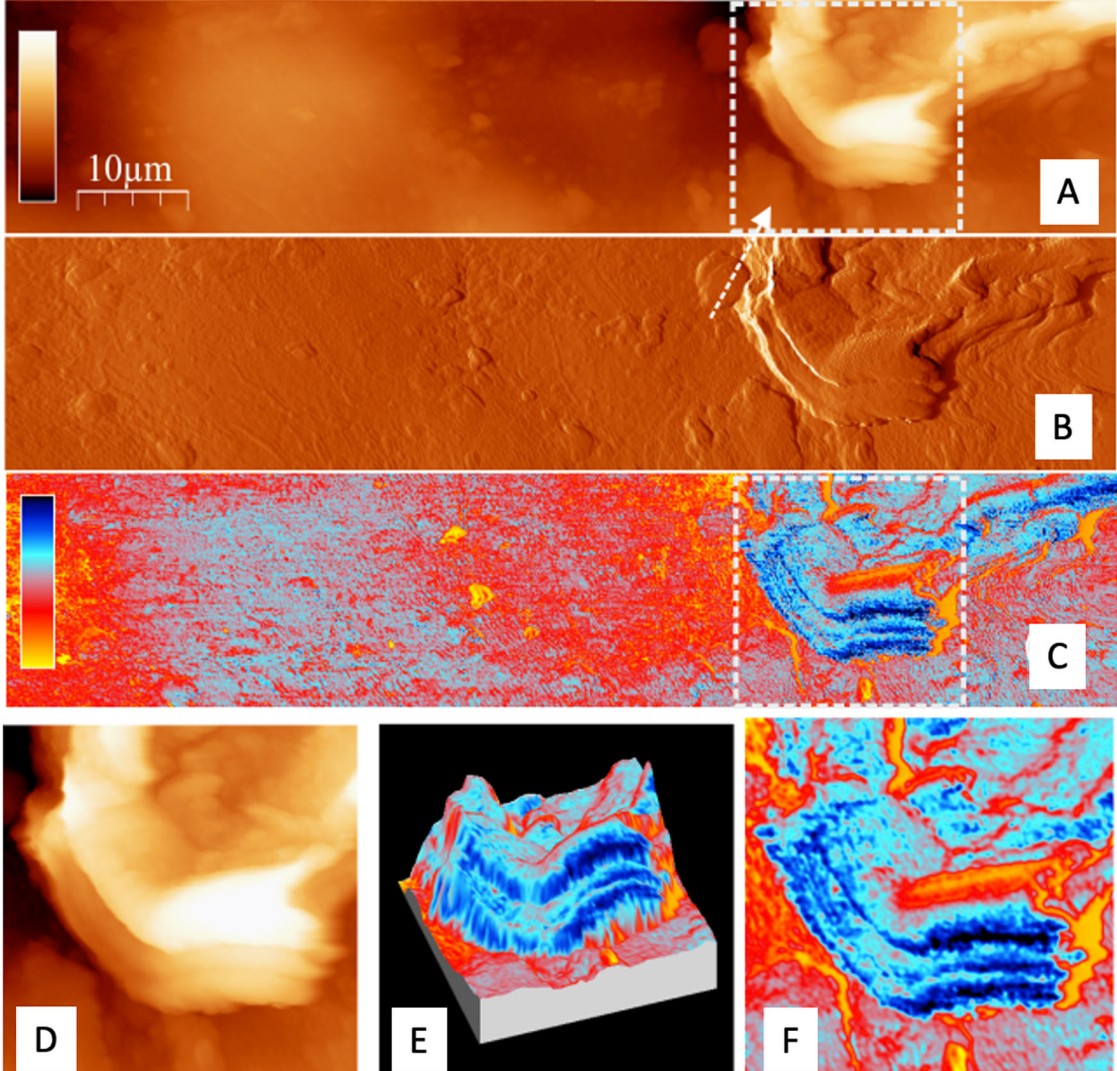

**Fig. 6 | AFM images of the adaxial side of an olive leaf, showing the epicuticular wax surface, and the edge of a trichrome.** Upper 3 rows (**A–C**): rectangular 50 µm × 10 µm region showing the topography (top rectangular image, A, total color scale corresponding to 5 µm height) and oscillation amplitude (middle rectangular image, **B**, a.u.), corresponding to the error signal that is kept constant by the feedback. This image enhances the contrast of edges and small features. Lower rectangular image, C: frequency shift data, corresponding to the nanoscale wettability, total color scale corresponds to 400 Hz, with blue corresponding to more wetting and yellow-red to less wetting (see color scale left). Lowest row: zoom of the region around the trichrome marked in the topography and the frequency shift image. **D** corresponds to the topography (lateral size 10 µm × 10 µm, total color scale 4 µm), the right image, **F** reflects frequency shift data and thus chemical wettability. **E** shows a composed, 3-dimensional image combining the topography and the chemical wettability. The vertical height of this surface map shows the morphology of the leaf surface, while the color superimposed on this surface map represents the nanoscale wetting properties, represented by the color of each point of the surface; blue: more hydrophilic, towards red: more hydrophobic.

feasible to evaluate if some of the abaxial trichomes were also shed due to the occurrence of trichome layers in this highly pubescent leaf side, it became clear that trichomes were damaged specially toward their edge (see arrows in Fig. 7C). We could not appreciate any structural damage in the adaxial surface of epidermal pavement cells which is densely covered with epicuticular waxes (Fig. 7A), while the abaxial epidermal cell surface which is also protected by pavement cells in addition to stomata, remained hidden underneath the dense abaxial trichome indumentum also after enzyme treatment (Fig. 7B, D).

The 6-months-old olive leaves analyzed had a significantly higher wax coverage in the adaxial ($227 \pm 16$ µg cm$^{-2}$) compared to the abaxial ($135 \pm 7$ µg cm$^{-2}$) leaf side, which is protected by a dense indumentum of peltate trichomes (Fig. 1A versus D). It must be noted that we generally failed to observe a distinct epicuticular wax layer onto the cuticle of trichomes in TEM leaf cross-sections (Figs. 2 and 3).

## Discussion

In this study, we analyzed the surface features and macroscopic wettability of the upper and lower side of Arbequina olive leaves, for further gaining insight into the chemical composition and structure with finer microscopic approaches including AFM and TEM. Olive is a major fruit crop grown in many arid and semi-arid areas of the world which is adapted to stress factors such as drought and high irradiation[40–43]. It is hence likely that olive leaves may have special features for improving atmospheric water-leaf surface interactions, and this hypothesis led us to carried out a detailed characterization of their wettability and physico-chemical properties. In addition, olive leaf surfaces are challenging owing to the occurrence of multicellular, peltate trichomes that form a dense indumentum on the lower side, and are scattered, with a predominance of epicuticular waxes in the upper side. Olive leaf trichomes provide UV-B protection[40,41,54] and are producers of enzymes associated with phenolic compounds in response to biotic and abiotic stress

**Fig. 7 | Effect of enzymatic treatment on olive leaf surface integrity. A** Adaxial leaf epidermis of an untreated leaf (8 days in water), **B** Abaxial epidermis of an untreated leaf. **C** Adaxial leaf surface and **D** Abaxial leaf surface after treatment with polysaccharide-degrading enzymes (8 days in an aqueous solution containing 4% cellulase and 4% pectinase enzymes). Arrows in **C** and **D** indicate trichrome scars (shed trichomes) and/or clearly degraded trichome areas.

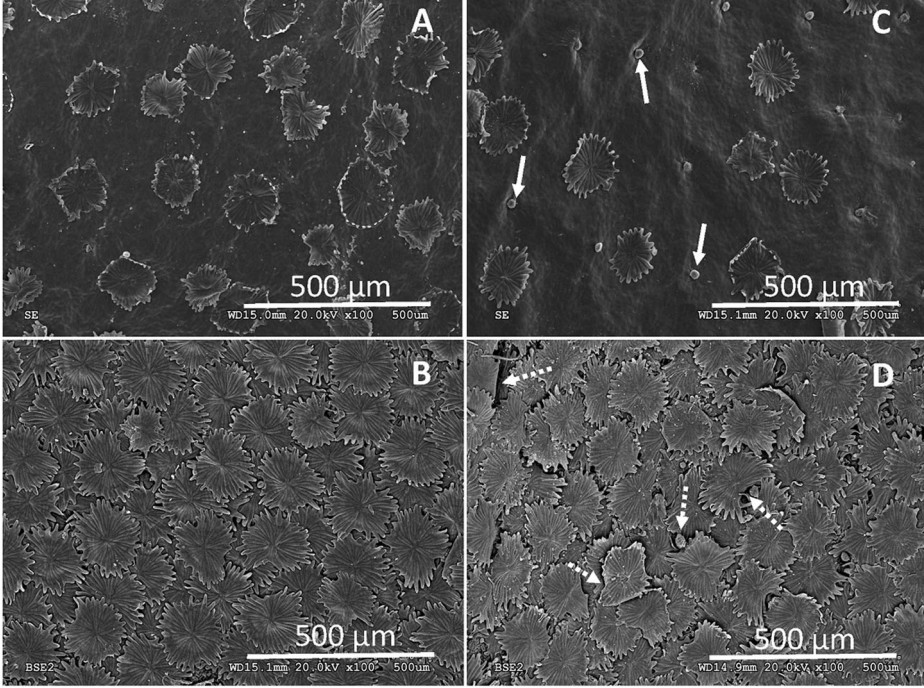

factors[55]. Nevertheless, it is clear that they may provide particular properties to leaf surfaces in terms of wettability and potential atmospheric water capture, as proposed by Bei et al. [31] for *Eleagnus angustifolia* leaves. Some studies analyzed the anatomical features of olive leaves[42] and the role of trichomes on stomatal conductance[42]. On the other hand, the chemical composition of Arbequina leaf cuticles was assessed by Huang et al. [56] which were reported to have *n*-alkanes as predominant very-long-chain acyclic wax compounds (78% of the total waxes). Hence, olive leaves surfaces are principally covered with epicuticular waxes in the adaxial (upper side) with decreasing, scattered trichome numbers as they become older[54]. By contrast, the lower (abaxial) leaf side of leaves older than 2 months is protected by a dense indumentum formed by various layers of peltate trichomes[54].

By implementing several techniques with a gradual degree of detail, we were able to gain evidence for the occurrence of polysaccharides at the surface of a plant trichome which significantly influence the wettability properties of olive leaves as described below.

Olive leaf surface wettability and surface free energy were estimated by measuring contact angles of drops of liquids with variable dispersive and non-dispersive components. following the Lifshitz-van der Waals-acid-base method, or van Oss, Good and Chaudhury method[47,57,58]. The surface free energy ($\gamma_s$) of intact plant organs and enzymatically isolated leaf cuticles has been assessed in some investigations[9,25,27,37,38,59,60].

Since many plant surfaces are glabrous (i.e., do not have trichomes) and are covered with relatively smooth epicuticular waxes such as *Ficus elastica* leaf or pepper fruit[48], the $\gamma_s$ values are often within the range described for paraffin waxes which vary between 23 and 35 mJ m$^{-2}$ (refs. [61,62]). For olive leaves, we determined $\gamma_s$ values of 31 and 37 mJ m$^{-2}$ for the predominantly waxy upper leaf side compared to the fully trichome-covered lower leaf side, respectively. Hence, both leaf surfaces were wettable by water and performed as is they were rather flat and smooth, including also the lower leaf side. By contrast, Leon and Bukovac[39] reported higher water contact angles of 106.2 and 125.4 ° for the adaxial and abaxial sider of 'Manzanillo' olive leaves which also have peltalte trichomes. The lower wettability of such 'Manzanillo' olive leaves may be for example associated with leaf age, measuring procedures, environmental conditions affecting leaf ontogeny and variety analyzed. Indeed, different cultivars of the same species may actually have strikingly variable leaf wettability properties, as recently shown by Henningsen et al. [63,64] for 4 corn varieties.

The abaxial side however had a higher $\gamma_s$ value and a higher polarity. The significantly higher degree of polarity of the abaxial (lower) leaf side (25%) compared to the more waxy adaxial (upper) side (8% polarity), suggests the occurrence of non-dispersive, hydrophilic material likely present in small quantities in the lower leaf side which also raised the $\gamma_s$ value, as reported for pubescent corn varieties[65]. It must be noted that the polarity of the adaxial side of younger leaves was also higher (23%) and decreased (to 8%) with leaf age (See Table S2). The water contact angles, $\gamma_s$ and $\delta$ values estimated for adaxial olive leaf surfaces are similar to those of mature *Quercus ilex* leaves[37]. However, differences between both species were recorded for the pubescent lower leaf side, olive being in the wettable threshold (90° for water) and having a slightly higher $\gamma_s$ of 37 mJ m$^{-2}$, compared to the water-repellent and a low energy surface of *Q. ilex* abaxial leaf surfaces[37].

The solubility parameter ($\delta$) of olive leaf surfaces was also determined, values being in the range of pure wax compounds (i.e., 15-16 MJ$^{1/2}$ m$^{-3/2}$)[65] in the upper side. The slightly higher $\delta$ values estimated for the lower leaf side are likely due to the occurrence of hydrophilic compounds in trichome surfaces as discussed below.

On the other hand, immersing young leaves (having higher trichome densities in the upper side compared to older leaves) in cell wall degrading enzymes (i.e., cellulase and pectinase) led to trichome surface damage as also reported for *Quercus ilex*, adaxial trichomes of juvenile leaves[37]. This indicates that trichome surfaces are not impermeable and have cellulose and/or pectin polymers exposed at the surface level which were degraded by the enzymes. The contrasting properties conferred by trichomes as shown here for olive versus *e.g., Q. ilex* leaf surfaces[37] provide evidence for variable chemical and structural nature of plant trichomes which should be explored in future investigations.

In general, both olive leaf surfaces are wettable by water and have a major degree of water drop adhesion (see Fig. 1). Being rather smooth and having a high wax coverage, it could be theoretically expected[66] that the adaxial leaf side may had higher equilibrium water contact angles of ~100°, but this was not the case as also reported for the *Q. ilex* adaxial leaf side[37].

AFM is a powerful tool for material surface topography and composition characterization, which has been recently applied for estimating the nano-wetting properties of rose petals having high water contact angles together with drop adhesion[19]. Indeed, the study by Almonte et al. [19] showed

that an aerial plant organ covered with an epidermis may have hydrophilic areas at the very surface level, despite the occurrence of cell wall poly-saccharides has been suggested in some investigations devoted to analyze the absorption of water and solutes by plant leaves[43,67–70]. Using olive leaf as model pubescent plant material, we here demonstrate that trichome sur-faces can be chemically heterogeneous at the nano-scale as enabled by the use of AFM techniques[19].

The combination of OM and AFM topographic images (Fig. 4) enables the accurate selection of AFM measuring areas which proved key for finding the scattered trichomes occurring in the upper side of the ~6 months old, olive leaves analyzed. This also helped us measure the edge of a trichome together with the properties of the background wax material covering the epidermal pavement cells of the upper leaf side (Fig. 7). By AFM, we esti-mated the different wetting and adhesion properties of the trichome versus the surrounding epicuticular wax surface of the upper leaf side, which supports our TEM and SEM observations of intact and polysaccharide-degrading enzyme treatments of adaxial leaf surfaces (Figs. 2 and 7, respectively). The chemical heterogeneity and major adhesion of water to trichome surfaces is likely associated with the occurrence of hydrophilic areas (Figs. 4–7) that may correspond with the zones attacked by polysaccharide-degrading enzymes. However, trichome damage caused by 1-week enzymatic treatment is shown macroscopically in SEM micrographs (Fig. 7), and it is likely that polysaccharide cell wall degradation by the enzymes began at this nano-scale hydrophilic surface zones, with damage progressing over time.

Regarding cuticle ultra-structure, it is remarkable that adaxial, epi-dermal pavement cells are covered with a thick, Type 6 cuticle[7,50] with a significant epicuticular wax layer. Abaxial pavement cells also have a Type 6 cuticle which is however thinner and rough due to the occurrence of cuti-cular folds that likely yield this surface were stomata are present, more unwettable than the upper leaf side, as described for the lower side of a rose petal[19]. By contrast, the cuticle of trichomes is largely reticulate and belongs to Type 4[7,50]. Hence, our study shows that variable cuticle, inner and surface patterns can be distinguished in different zones which implies that leaf surfaces mat be more complex than currently believed.

While the lower leaf side of Arbequina olive leaves is fully covered with trichomes, the upper surface is dominated by a thick layer of epicuticular waxes which manly correspond to n-alkanes[56]. Such waxes are fully apolar and can be expected to have poor non-dispersive and H-bonding interaction with water[65]. Indeed, we observed an increased adhesion and wettability of trichome surfaces compared to the background epicuticular waxes covering most of the upper leaf side, which provides evidence for the robustness of the AFM methodology recently introduced by J. Colchero[19].

Peltate trichomes have a major presence in Arbequina olive leaves and are totally associated with the surface features of the abaxial leaf side which is fully covered with these epidermal, multicellular structures. Despite the upper leaf side of the 6-months-old leaves analyzed in this investigation is dominated by epicuticular waxes (manly n-alkanes[56]) which are rather apolar and may lead to advancing contact angles between 100 and 110 ° as described for rather smooth paraffin surfaces[66], we determined mean advancing angles of 85° which can be related to the higher wettability of the trichomes that despite being scattered, change the overall wettability of the upper olive leaf surface compared to a fully waxy material as described by e.g., Kamusewitz et al. [66]. Indeed, in spite of being different regarding tri-chome densities, the upper and lower leaf side did not greatly differ in terms of water wettability and a high contact angle hysteresis. It must be noted that trichomes are composed by ca. 70 µm long cells. Their large size and rather smooth surface, yield the overall surface flat and wettable, despite the occurrence of trichome layers on the lower leaf side. This was initially surprising and contra-intuitive for us, who potentially expected surfaces to be more unwettable by water. The detection of nano-scale hydrophilic areas in trichome surfaces and maybe trichome scars, may be related to higher wettability of the upper leaf side despite such surface is largely covered with apolar paraffin waxes as also determined for Q. ilex leaves[37]. However, more detailed trichome and trichome scar surface analyses should be performed

in future investigations for better understanding the wettability of leaves having epidermal outgrowths.

Our results show that trichomes provide water adhesion to olive leaves and may facilitate the retention and even the foliar absorption of surface-deposited water associated with dew, rain or fog, as described in some studies[26,30,31]. While a clear UV-B protection role has been described for olive leaf trichomes and no function in preventing water loss has been demon-strated to date[40–42,45], we cannot discard the possibility that may be functional for retaining surface-deposited water as suggested for E. angustifolia, peltate leaf trichomes[31]. This phenomenon may contribute even to a low extent, to olive tree water economy during drought spells, when it may rain and dew may form overnight, as observed during great periods of the growing season at least in Mediterranean areas[37]. The beneficial contribution of trichomes as water retainers in leaves has been also documented in other plant species[31,71,72].

The high wettability of adaxial, olive leaf surfaces will lead to film water condensation/ leaf wetting and to the potential run off of rain and con-densed water to the roots of olive trees. Leaf water adhesion to trichomes as shown in our investigation, may also decrease leaf temperature and create a more favorable environment for physiological functioning. However, fur-ther trials carried out with fine methodologies for detecting or modeling water transport phenomena in olive leaves should be carried out for clar-ifying to potential role of trichomes in leaf water balance. The occurrence of nano-scale, hydrophilic-hydrophobic areas in olive leaf trichomes may also have implications for insect and pathogen surface interactions[73,74] that should be assessed in future investigations.

## Methods
### Plant material
Experiments were generally developed with 6-month old, healthy leaves collected from a commercial plantation (Viña Elena Jumilla, Murcia (38° 25'5.5" N – 1° 15'50.3" W)) of 9-year-old olive-trees (Olea europaea L. var. Arbequina) grown with a frame of 3.5 × 1.5 m. The features of young (approximately 3-months old) versus old (1 to 2 years old) leaves were evaluated in terms of pseudo-equilibrium contact angles (leaves of both age) and polysaccharide-degrading enzyme attack (only assessed with young leaves) as described below.

The upper (adaxial) and lower (abaxial) surface of intact, fresh (i.e., with no further preparation) olive leaves was directly analyzed by AFM and SEM. Olive shoots were collected in the field, rapidly wrapping them in wet paper after sectioning and keeping in plastic bags at low temperature during transport to the laboratory. The stem of the shoots was then immersed in a beaker full of tap water and kept in the fridge at approximately 6 °C (from 1 to 3 days) for the development of measurements.

### Scanning Electron Microscopy
The adaxial and abaxial side of 6-month old olive leaves was directly observed by field emission scanning electron microscopy (FESEM, SIGMA 300 VP, Zeiss, Germany, coupled to an X-ray, EDX detector, Universidad Miguel Hernández, Elche, Spain) and variable pressure SEM (Hitachi S-3000N, Hitachi High-Tech, Japan; Universidad Autónoma de Madrid, Spain) at low vacuum, after gold coating (Quorum Q150T-S sputter, Quorum, United Kingdom) and also without metal sputtering. Fresh leaf sections of approximately 0.5 cm2 were cut and surfaces were observed at 10.0 kV and approximately 8.5 mm working distance[75].

### Observation of leaf cross-sections.
For optical microscopy (OM) and transmission electron microscopy (TEM) examination, 6-month-old olive leaves were cut into 4 mm$^{-2}$ pieces and fixed in glutaraldehyde (2.5%) and paraformaldehyde (4%) (both from Electron Microscopy Sciences (EMS), USA) for 3 h at 4°C. Samples were rinsed in cold phosphate buffer (7.2 pH) 4 times for 6 h and were kept at 4 °C for 12 h. Leaf pieces were post-fixed in a 1:1 aqueous solution of osmium tetroxide (2%, TAAB Laboratories, United Kingdom) and potassium ferrocyanide (3%, Sigma-Aldrich, Germany) for 1.5 h. They were rinsed in distilled

water (x3), dehydrated in an acetone series (30, 50, 70, 80, 90, 95 and 100%; x2, 15 min each concentration) and embedded in acetone-Spurr's resin (TAAB Laboratories) mixtures (3:1 for 2 h, 1:1 for 2 h, 1:3 for 3 h (v:v)) and finally in pure resin, keeping samples at room temperature. After 12 h, olive leaf sections were placed in blocks which were filled with pure resin, before incubation at 70°C for 3 days until complete resin polymerization. Ultra-thin sections (obtained with a Leica Ultracut E, Leica Microsystems, Germany) were cut, mounted on nickel grids and post-stained with Reynolds lead citrate (EMS) for 5 min before microscopic observation. Leaf cross-sections were observed with a Jeol 1010 TEM (Jeol Ltd., at 80 kV) equipped with a CCD Megaview camera (National Electron Microscopy Centre, Complutense University of Madrid, Spain). On the other hand, for leaf OM analysis, semi-thin cross-sections were cut, mounted in microscope slides and stained with Toluidine Blue, before observation with an epifluorecence microscope (Axioplan-2, Zeiss, Germany).

Trichome features on adaxial and abaxial surface of approx. 2-, 6- and 12- to 24-month-old leaves were grossly assessed on SEM micrographs (10 leaves and 2 different sections per leaf). The density (or frequency expressed as number of trichomes per unit area, $N = 20$ different images), length and diameter ($N = 120$ trichomes) of the multicellular, peltate trichomes occurring on 6-month-old leaves were determined by SEM image analysis using ImageJ program (National Institutes of Health, Bethesda, Maryland, USA)[44].

### Contact angle measurements

Pseudo-equilibrium contact angles of drops ($N = 30$) of deionized water, glycerol (99% purity, Sigma-Aldrich) and diiodomethane (99% purity, Sigma-Aldrich) were determined with a Drop Shape Analysis System (DSA 100, Krüss, Germany) at room temperature (~25 °C), making sure liquid drops were stable before measurement. For preserving surface integrity, 2, 6 and 12–24 month-old olive leaves, collected from shoots were kept in water at 6 °C and immediately excised before measurement. After removing the midrib and margin, leaf sections of ~$2 \times 0.5$ cm$^2$ were cut with a scalpel and mounted on a microscope slide with double-sided adhesive tape. A dosing system holding a 1 ml syringe with a 0.5 mm diameter needle was used for supplying approximately 2 µl drops of each liquid onto adaxial and abaxial olive leaf surface. Side-view images of stable drops were taken and contact angles were automatically calculated using the tangent method ($N = 30$). The Lifshitz-van der Waals/ acid-base method or van Oss, Good, and Chaudhury method[47,57,58] referred to as the 3-Liquids Method, for simplicity) was used for estimating the surface free energy of olive leaf surfaces. For calculating the total surface tension ($\gamma_l$) and surface tension components ($\gamma_l^{LW}$, Lifshitz–van der Waals; and Acid-base positive ($\gamma_l^+$) and negative ($\gamma_l^-$) components) of adaxial and abaxial olive leaf surfaces by this procedure, the following liquid mean values were employed[48]: $\gamma_l = 72.80$ mJ m$^{-2}$, $\gamma_l^{LW} = 21.80$ mJ m$^{-2}$, $\gamma_l^+ = \gamma_l^- = 25.50$ mJ m$^{-2}$ for water, $\gamma_l = 63.70$ mJ m$^{-2}$, $\gamma_l^{LW} = 33.63$ mJ m$^{-2}$, $\gamma_l^+ = 8.41$ mJ m$^{-2}$, $\gamma_l^- = 31.16$ mJ m$^{-2}$ for glycerol and $\gamma_l = \gamma_l^{LW} = 50.80$ mJ m$^{-2}$, $\gamma_l^+ = 0.56$ mJ m$^{-2}$, $\gamma_l^- = 0$ mJ m$^{-2}$ for diiodomethane.

The polarity of leaf surfaces was estimated as a percentage of the acid-base component ($\gamma_s^{AB}$) divided by the total surface free energy ($\gamma_s$). In addition, advancing ($\vartheta_{adv}$) and receding ($\vartheta_{rec}$) contact angles ($N = 20$) of water onto the abaxial and abaxial surfaces of 6-month old olive leaves were estimated[23,76]. For determining $\vartheta_{adv}$ onto olive leaf surfaces, drops of increased volume were formed at the end of a syringe needle. Drops adhered to both olive leaf surfaces and did not slide when moving the samples in the horizontal plane. Drop volumes were gradually increased to reach up to 9 to 10 µl, when motion of the contact line was observed to occur and maximum $\vartheta_{adv}$ values were recorded. Receding contact angles ($\vartheta_{rec}$) were measured after reducing drop volumes to approximately 1–2 µl water (drops remained attached to the needle). The water contact angle hysteresis ($\Delta\vartheta_{hy}$) of adaxial and abaxial olive leaf surfaces was estimated as the difference between $\vartheta_{adv}$ and $\vartheta_{rec}$.

In this study, we refer to "wettable" when the surface has a contact angle $\vartheta_{cont} < 90$, that is, $\cos(\vartheta_{cont}) = \Delta\gamma/\gamma_L = 0$, where $\Delta\gamma$ is the difference of surface energy between the uncovered surface (described by $\gamma_S$) and the surface covered by a liquid (described by $\gamma_{SL}$), and $\gamma_L$ is the surface energy of the liquid.

### Wax extraction

For soluble cuticular lipid (wax) extraction, 3 groups of 30 healthy leaves of similar age (approx. 6 months-old) were selected and scanned for measuring their total area by image analysis (ImageJ[44]). Leaves were clamped from the petiole with forceps and the adaxial or abaxial side of each leaf washed by carefully applying chloroform (Sigma-Aldrich) flushes with a Pasteur pipette (6 flushes of ca. 1.7 ml each per leaf side). Chloroform extracts were filtered with Whatman filter paper Grade 1 and were kept in glass beakers in a laboratory hood (at 25 °C with aeration), until a substantial amount of chloroform had evaporated. When approximately 10 ml of chloroform extract remained in the beakers, the extract was carefully deposited onto glass watches which had been previous weighted using a precision balance. The amount of waxes of each leaf side was determined gravimetrically and expressed as weight per unit leaf area ($N = 3$).

### Leaf enzymatic treatment

For attempting to degrade cell wall polysaccharides potentially occurring at the leaf surface, approximately 3-month old olive leaves ($N = 10$) which had higher trichome densities in the adaxial side compared to older leaves, were collected and immersed in a solution containing 4% cellulase (Novozymes, Bagsvared, Denmark), 4% pectinase (Novozymes), and 2 mM sodium azide (Sigma-Aldrich)[37]. As control treatment, an additional group of similar leaves ($N = 10$) were immersed in a 2 mM sodium azide solution. After 8 days, leaves were taken out from the enzymatic solution and were air-dried prior to direct SEM observation, as described above.

### Atomic force microscopy

The nano-scale topography and wettability of 6-month-old olive leaf surfaces was estimated by AFM. Data acquisition was in Dynamic Atomic Force Microscopy (DAFM), using a Nanotec Electronica AFM system (Madrid, Spain) with a phase-locked loop board (PLL, bandwidth ~2 kHz). Images were acquired using the amplitude as signal for the principal feedback channel (in the so-called Amplitude Modulation dynamic mode; AM-DAFM) at (relatively) large oscillation amplitude ($a_{free} \approx 25$ nm), but low amplitude reduction set-point ($a_{set}/a_{free} \approx 0.9$-$0.8$), to keep tip-sample interaction in the attractive, non-contact regime. Silicon tips with a nominal force constant of 2 N/m were used. The nominal radius of the tip apex of these probes is specified as 15 nm by the manufacturer (Olympus AC240TS). In our experiments, the oscillation amplitude of the cantilever (error signal) is kept constant by the AFM feedback loop, and the frequency shift $\delta\nu = \nu(X,Y) - \nu_0$ ($\nu_0$: free resonance frequency) of the tip-sample resonance frequency was determined using the PLL system of our electronics. This frequency shift $\delta\nu$ was acquired as additional channel of information related to the chemical composition of the sample[19]. WSxM software was used for image processing[77]. Typically, a plane filter was applied to topography images; no filter is applied to the frequency images[19].

### Statistics and reproducibility

The sample size for the analyses described above varied between 5 to 30 leaf repetitions collected from 10 different Arbequina olive tree clones. For TEM and AFM, sections of 5 different leaves were analyzed by observing different leaf areas per leaf piece.

Contact angles, trichome lengths and surface areas were analyzed by ANOVA. Tukey's honest significance tests (HDS) were carried out to estimate differences between factors when F-values were significant ($p < 0.05$). Statistical analyses were carried out using SPSS 23.0. The number of biological replicates and p-value are provided in table legends and indicated in the Methods section also for microscopic observations.

## Reporting summary

Further information on research design is available in the Nature Portfolio Reporting Summary linked to this article.

## Data availability

All data supporting the findings of this study are available within the paper and as Supplementary Information. All other data are available from the corresponding author on reasonable request.

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

## Acknowledgements

This work was supported by PID2019-104272RB-C52, PID2022-139191OB-C31, in addition to TED2021-130830B-C41 and TED2021-130830B-C42, which are financed by MCIN/AEI/10.13039/501100011033 and European Union NextGenerationEU/PRTR funds. G.S.A. is supported by a María Zambrano contract of "Programa de Ayudas para la Recualificación del Sistema Universitario Español". We want to thank Villa Elena for their support to carry out trials in their commercial olive plantations. Thanks to Novozymes for providing enzyme samples for experimental purposes.

## Author contributions

Conceptualization: V.F., H.A.B. and J.C.; Investigation: V.F., L.F.A., H.A.B., A.G., G.S. and J.C.; Data analysis: H.A.B., L.F.A. and J.C.; Writing – original draft: V.F.; Writing – review & editing: V.F., L.F.A., H.A.B., A.G., G.S. and J.C.

## Competing interests

The authors declare no competing interests.
