## [Peer Review File · Communications Biology]

Reviewers' comments:

Reviewer #1 (Remarks to the Author):

Dear Author,

I have now carefully read the manuscript entitled Olive leaf wettability: from nanoscale to macroscale, focus on trichomes. This study's main objective was characterising the olive leaf's surface as a plant material. The manuscript is well-written and contains an interesting wide range of microscopic examination results, which deserve publication in a biological journal. However, I think some comments should be considered before publication. Therefore, my decision is a revision.

1. The authors use a pseudo-equilibrium contact angle but do not explain which angle value they consider equilibrium.

2. To check the correctness of the calculations of the surface energy of olive leaves, it is required to know the values of both acid-base components, i.e. the electron acceptor (positive) γ^+ and the electron donor (negative) γ^- , but the authors do not provide these values. I believe these values need to be included in the revised manuscript.

3. I need clarification about the age of the leaves taken for experiments. Sometimes, the authors use the terms old and young leaves, but they need to write how they examined the age of the leaves or which leaves they consider young and which old.

4. In the methods section, there needs to be more information about the method for determining the surface energy of olive leaves. I don't think there's any reference to relevant literature. Khayet and Fernández 2015 are not the authors of the method, so appropriate citations should be included in the methods section. The part that should be here is in the Results section (Lines 122-125 or 130-133). Moreover, the authors did not measure the values of liquid surface tension; they only used literature values. It is better to provide the original literature (L. 649).

5. The study's first hypothesis that the abaxial and adaxial parts of leaves have different wettability is evident, especially since such studies have already been presented in numerous kinds of literature, for example, Brewer and Nunez 2007; Holder 2007; Papierowska et al. 2018.

Brewer, C. A., & Nunez, C. I. (2007). Patterns of leaf wettability along an extreme moisture gradient in western Patagonia, Argentina. *International Journal of Plant Sciences*, 168(5), 555-562.

Holder, C. D. (2007). Leaf water repellency as an adaptation to tropical montane cloud forest environments. *Biotropica*, 39(6), 767-770.

Papierowska, E., Szporak-Wasilewska, S., Szewińska, J., Szatyłowicz, J., Debaene, G., & Utratna, M. (2018). Contact angle measurements and water drop behavior on leaf surface for several deciduous shrub and tree species from a temperate zone. *Trees*, 32, 1253-1266.

6. Leon and Bukovac (1978) investigated the wettability of olive leaves and presented the results of contact angle measurements. They obtained higher contact angle values in both adaxial and abaxial leaf parts (106 and 125 degrees, respectively) than presented in this manuscript. Could you explain why the results are so different? Do you think it depends on the conditions in which they grow? If so, add information about the location, climatic and meteorological conditions, etc.

It would be interesting if the discussion section included comparing the obtained contact angle and surface free energy results with values presented in the literature.

I strongly recommend studying the literature; then, there will be less self-citation, especially in the Introduction section.

Leon, J. M., & Bukovac, M. J. (1978). Cuticle Development and Surface Morphology of Olive Leaves with Reference to Penetration of Foliar-applied Chemicals¹. *Journal of the American Society for Horticultural Science*, 103(4), 465-472.

7. You should emphasise the separation of leaf surface structure into macrostructure and

microstructure because of some inaccuracies. The author wrote that the surface roughness is low (L.130), and then they show the surface topography in Fig. 6e.

Minor comments:

1. In the Abstract, could you explain what key functional properties you have in mind? Maybe it's worth mentioning which ones. (Lines 40-42)
2. Add Johnson, 1975, to the literature list and check the correctness of this sentence (Lines 79-80).
3. Move lines 122-125 and 130-133 from the Results section to the Methods section.
4. Add Jeffree, 2006, to the literature list (Lines 156, 162-163).
5. In line 198, you wrote that the yellow-red colour relates to rather hydrophobic, which sounds like you're unsure about. Am I right?
6. Use space between values and units (Lines 190-191).
7. At the beginning of the Discussion section, you use twice the same information about UV-B protection (L. 261 and L. 267).
8. Add Karabourniotis et al. 1995, to the literature list (L. 268)
9. In the Method section, add information about how long the samples were kept in the fridge.
10. Add Khayet and Fernández 2015 to the literature list or change it to Fernández and Khayet 2015 if it is correct.
11. Always use a lower or upper case letter to indicate the number of samples (n = 20 L. 650; N = 30 L. 645).
12. The determination of surface free energy should be presented as a subsection.
13. Could you add the methodology of polarity calculation in the Methods section?
14. In Figs. 4, 5, and 6, please enlarge the scale bar because it is illegible.
15. In Table 2S, use capital letters in line captions.
16. In Fig. S2, add scale bar.

Reviewer #2 (Remarks to the Author):

The manuscript entitled "Olive leaf wettability: from nanoscale to macroscale with focus on trichomes" by Victoria Fernández and colleagues analyzes an important property of the leaf surface by different methods. The subject is interesting, but the manuscript needs to be improved to be suitable for publication.

Lines 33-34: the statement is too general to be included in the Abstract; previous papers reported the same distribution of trichomes - predominantly on the lower epidermis (Koudounas et al., 2015 - cited in References in this manuscript); do the authors know any example of *Olea* sp. where the trichomes are mostly located on the upper epidermis, in mature leaves?

Introduction - the first paragraph: the description of the structure of the epidermis and the cuticle is a bit simplistic, rather at the level of a university course than a scientific paper; I recommend the authors to describe in more detail the chemical composition of the cuticle because it has an essential role in the investigations carried out. The following papers can be used as references:

1. Li-Beisson, Y., Verdier, G., Xu, L., & Beisson, F. (2016). Cutin and suberin polyesters. *eLS*, 1-12.
2. Philippe, G., Sørensen, I., Jiao, C., Sun, X., Fei, Z., Domozych, D. S., & Rose, J. K. (2020). Cutin and suberin: assembly and origins of specialized lipidic cell wall scaffolds. *Current Opinion in Plant Biology*, 55, 11-20.

line 107: there is a discrepancy between the title of the chapter "Macroscopic olive leaf wettability" and the references to Figures 1A and D - obtained at SEM. These are not macroscopic observations!

lines 109-110: "contact angle measurements were measured" - repetition, please rephrase.

lines 112 and 117: how did you calculate the density of trichomes per mm²? How many fields were measured?

line 118: how did you estimate 9 ± 5 % trichome coverage?

line 144: "Olive leaf surface structure and chemical composition" - I did not find in the content references to the chemical composition of the leaf surface (the references in line 171 to the possibility of the presence of polysaccharides in the structure of the cuticle do not think that it can be considered chemical composition).

line 147: "Figures 2 to 4" - figure 4 is not obtained by TEM or light microscope (it is an AFM image).

line 148: "Toluidine (correct Toluidine) Blue which provides cell wall polysaccharides a blue color" - on the sections made on plant material included in resins, the color obtained for polysaccharides is purple; only on manually sectioned plant material, where the sections are thicker (usually over 50-60 μm), the obtained color is blue. How thick are your sections? I recommend the following paper as a reference:

1. Ribeiro, V. C., & Leitão, C. A. E. (2020). Utilisation of Toluidine blue O pH 4.0 and histochemical inferences in plant sections obtained by free-hand. *Protoplasma*, 257(3), 993-1008.

line 198: "No significant correlation" - how did you come to the conclusion that the correlation is not significant?

lines 237-239: "were immersed in solutions containing cellulose and pectin degrading enzymes (i.e., cellulase and pectinase, respectively) for 8 days and then observed potential surface changes by SEM. " - don't you consider that incubation in aqueous solution for 8 days (!!) of a plant material without fixation or preservation can alone induce significant structural changes? And were the control samples incubated in the same conditions (without enzymes)? Or were fresh leaf samples used as a control? In this case, a comparison cannot be made.

lines 387-388: "trichomes are composed by ca. 50 μm long, elongated cells and have a large size." - this statement appears in the Discussions, but in the Results there are no results of the measurements performed on the trichomes.

line 388: what does "macroscopic nature" mean regarding trichomes?

line 595: "eventually evaluated" - please specify exactly the investigations carried out.

lines 607-609: please specify exactly how the SEM images in the manuscript were made; did you use low vacuum with gold coated samples? What was the thickness of the gold layer applied? On lines 597-598 it is specified "The upper (adaxial) and lower (abaxial) surface of intact, fresh (i.e., with no further preparation) olive leaves was directly analyzed by AFM and SEM." Has coating/drying been achieved or not for SEM observations?

line 622: what were ultra-thin sections made with? With glass or diamond knives? The thickness of the sections must be approximated using the color scale available for ultramicrotomes.

line 632: where do the length and diameter of the peltate trichomes appear in the Results? It is necessary to detail how many measurements were made, from how many leaves the trichomes were analyzed and to present the concrete results obtained.

line 678: "were air-dried prior to direct SEM observation" - if these samples were air-dried (which is known to induce significant plant tissue artifacts) was the control sample dried or not? I recommend the paper below to see what changes can be induced by air drying in plant samples for SEM:

1. Bhattacharya, R., Saha, S., Kostina, O., Muravnik, L., & Mitra, A. (2020). Replacing critical point drying with a low-cost chemical drying provides comparable surface image quality of glandular trichomes from leaves of *Millingtonia hortensis* L. f. in scanning electron micrograph. *Applied Microscopy*, 50(1), 1-6.

Reviewer #3 (Remarks to the Author):

In this work, Fernández et al. analyzed the wettability and macro- and micro-scale features of olive leaves. Combining different approaches, trichome surfaces have been found to be chemically (hydrophilic-hydrophobic) heterogeneous at the nano-scale. With it, the authors concluded that the chemical heterogeneity of trichomes may provide key functional properties to biological surfaces.

While this work has its merits and the authors provided adequate evidence for their statements, the writing can be confusing and challenging to comprehend. Despite my diligent efforts in re-reading the manuscript several times, I am left with lingering uncertainty about whether I may have inadvertently missed any critical details. The writing style hiding crucial details among expansive background information did not work in my favor.

Since trichome is the correct botany term it should be used throughout the text instead of “hair”.

Line 56 – “aprovide”?

Lines 61 – 63 – Which are these “bioinspired materials”? Which “major functional roles” can the wetting properties play? It would be interesting to explore the reason for studying wettability using olive more. For me, this information is not clear throughout the manuscript.

Lines 86 – 87 – Which are the compounds that absorb the UV radiation?

Line 109 – “static (with liquids of different nature)” Describe the liquids used in the experiment and why.

Line 174 – AFM must be defined in the first appearance in the text.

Line 242 – “adaxial trichomes” maybe the correct term here is abaxial trichomes.

Fig 2 – “LT, abaxial leaf surface.” I think that should be “abaxial leaf surface trichome,” right?; Although the abbreviations supposedly point to each of the structures, it would be interesting to delimit/outline areas where each of the structures shown is located. It is worth remembering that microscopy images are quite difficult for readers who are not used to them.

Fig 3 - The assembly of this figure needs to be improved. The images are larger than the frames; lines overlap previous elements of the pictures, there is no standardization of the position of the labels, etc.

Fig 4 – 6- I think the legends and the text associated with these pictures must be improved, considering people who have never worked with this technique.

Dear Editor,

We first want to wish you a happy and healthy 2024.

We thank you for the constructive and fair revision of our paper which definitely help us improve it quality.

We did our best to implement the suggestions of the referees in the new version of the draft and to reply to their comments in our response letter.

Thank you for giving us the opportunity to submit our revised manuscript to Communications Biology and best regards,

Victoria Fernández

Reviewer #1 (Remarks to the Author):

Dear Author,

I have now carefully read the manuscript entitled Olive leaf wettability: from nanoscale to macroscale, focus on trichomes. This study's main objective was characterising the olive leaf's surface as a plant material. The manuscript is well-written and contains an interesting wide range of microscopic examination results, which deserve publication in a biological journal. However, I think some comments should be considered before publication. Therefore, my decision is a revision.

Dear Reviewer 1, thank you for the revision of our manuscript and for your detailed comments that we attempted to implement in the revised version we now submitted. We will provide a detailed reply to your concerns below:

1. The authors use a pseudo-equilibrium contact angle but do not explain which angle value they consider equilibrium.

This is a good question, especially when dealing with plant materials having trichomes (e.g., many leaves from *Quercus* spp.) which in our experience, sometimes lead to initially high water and glycerol contact angles (CA) which decrease sometimes quite significantly after few seconds. We took the CA values after depositing drops and noting that they were stable (in general, we took the values approx. 10 sec after the deposition of the water and glycerol drops). For estimating pseudo-equilibrium CAs of any liquid it is important to measure drops which are static and we always check their performance when measuring a new leaf sample. For instance, measuring glycerol CAs with the abaxial side of *Quercus ilex* leaves required several minutes to reach stable drops (see Fernández et al. 2014. *Plant Physiol.*, 166(1), 168-180). The procedure described for measuring pseudo-equilibrium contact angles are standard, but in our opinion, plant materials are quite tricky and require special attention for analysing their wettability properly. We now made a remark on this on the Materials and methods section for the sake of clarity.

2. To check the correctness of the calculations of the surface energy of olive leaves, it is required to know the values of both acid-base components, i.e. the electron acceptor (positive) γ^+ and the electron donor (negative) γ^- , but the authors do not provide these values. I believe these values need to be included in the revised manuscript.

Thank you for your comment. You are fully right regarding the calculation of the acid-base surface free energy (SFE) component, but we avoided to provide them not to come up with a larger table 2. We now provide such values as requested.

3. I need clarification about the age of the leaves taken for experiments. Sometimes, the

authors use the terms old and young leaves, but they need to write how they examined the age of the leaves or which leaves they consider young and which old.

Thank you for your comment. We refer to approx. 3 months-old leaves as young and >1 year-old as "old"/mature leaves, as described in the first paragraph of the Materials and Methods section. We made efforts to clarify the age of leaves specially in the Materials and Methods section, but most measurements correspond to 6 months-old leaves.

4. In the methods section, there needs to be more information about the method for determining the surface energy of olive leaves. I don't think there's any reference to relevant literature. Khayet and Fernández 2015 are not the authors of the method, so appropriate citations should be included in the methods section. The part that should be here is in the Results section (Lines 122-125 or 130-133). Moreover, the authors did not measure the values of liquid surface tension; they only used literature values. It is better to provide the original literature (L. 649).

We tried to follow your recommendation and moved some lines to the Materials and Methods section. We avoided to include all the references of the 3-liquids method in the draft for reducing the length of the text and aware that the liquid surface tension values employed are means of all the reference data we were able to gather from the literature (Fernández and Khayet. 2015). When we wrote this paper, I grew petrified about the complexity to determine the surface tension and surface tension components of (reference) liquids and the variability of reported for some of them, measured by appropriate physico-chemical procedures (e.g., glycerol). Hence, we normally use the here described mean values for the 3 reference liquids we normally assess (i.e., water, glycerol and diiodomethane) and believe it is not suitable (for the sake of not coming up with a lengthy paper) to provide all the references we quoted when developing the 2015 methodological study.

On the other hand and following your suggestion, a sentence was included to describe the SFE calculation method used, including the related quotations.

5. The study's first hypothesis that the abaxial and adaxial parts of leaves have different wettability is evident, especially since such studies have already been presented in numerous kinds of literature, for example, Brewer and Nunez 2007; Holder 2007; Papierowska et al. 2018.

Brewer, C. A., & Nunez, C. I. (2007). Patterns of leaf wettability along an extreme moisture gradient in western Patagonia, Argentina. *International Journal of Plant Sciences*, 168(5), 555-562.

Holder, C. D. (2007). Leaf water repellency as an adaptation to tropical montane cloud forest environments. *Biotropica*, 39(6), 767-770.

Papierowska, E., Szporak-Wasilewska, S., Szewińska, J., Szatyłowicz, J., Debaene, G., & Utratna, M. (2018). Contact angle measurements and water drop behavior on leaf surface for several deciduous shrub and tree species from a temperate zone. *Trees*, 32, 1253-1266.

Thank you for your comments. We included all the references suggested. We now made reference to trichomes in the first hypothesis, but however, do not take anything for granted whenever we compare e.g., abaxial versus adaxial leaf surfaces of the same spp. Note that e.g., the water CA hysteresis of the upper and lower leaf side of 6-month old olive leaves is quite similar, despite there are only few peltate trichomes in the upper side which provided evidence that we should not trust our primary intuition after observing leaf surface features by e.g., scanning electron microscopy. We are missing the combined contribution of structure and chemical composition for leaf wettability and may be facing variable challenges we ignore by only trusting on the topography of the samples. This is what we are currently trying to estimate more in detail and why we posed such apparently simple first hypothesis.

6. Leon and Bukovac (1978) investigated the wettability of olive leaves and presented the results of contact angle measurements. They obtained higher contact angle values in both adaxial and abaxial leaf parts (106 and 125 degrees, respectively) than presented in this manuscript. Could you explain why the results are so different? Do you think it depends on the conditions in which they grow? If so, add information about the location, climatic and meteorological conditions, etc.

Dear referee, we were glad to read the paper by Leon and Bukovac (1978) on Manzanilla olive leaves which I did not come across before. I read it back and forth and failed to clearly understand how old were the leaves for which water CAs were measured. I also failed to be clear about the volume of drops (which is a key factor) they employed and what did they mean with "advancing contact angles". I also wonder if drops were actually determined in pseudo-equilibrium and were stable as the ones we measured.

Note that we have measured Arbequina leaves in different seasons and from different areas of East and Southern Spain and also from Talca (Chile), and we generally got mean CA values within a similar range, provided that leaves of comparable age were analysed (younger leaves are less wettable specially regarding their abaxial side).

We can hence not fully interpret the results reported by Leon and Bukovac (1978), but aware that they analysed var. Manzanilla and that we focussed on var. Arbequina. Working with 4 different corn varieties, we recently observed potentially major leaf wettability performance variations among cvs. Of the same species (see Henningsen et al. 2023. Leaf surface features of maize cultivars and response to foliar phosphorus application: effect of leaf stage and plant phosphorus status. *Physiologia Plantarum*, 175(6), e14093).

It is more than likely that the growing conditions also influence the properties of leaf surfaces but there is no way to safely compare leaf wettability results with the ones reported by Leon and Bukovac (1978) for the reasons explained above.

We now quoted with draft in the introduction and discussion emphasising we cannot safely compare our results with those of Manzanilla olive leaves.

On the other hand, the water retention data provided by the interesting Leon and Bukovac (1978) paper which are within a similar range for the adaxial and abaxial leaf side, somehow

support our water contact angle hysteresis measurements, but pose questions on the contact angle values reported in this investigation (105° vs 125 ° for the upper and lower side, respectively).

In summary, we did efforts to explain our results and we are not fully able to interpret the wettability measurements performed by Leon and Bukovac (1978) at least due to variety, measuring and environmental differences between our studies.

It would be interesting if the discussion section included comparing the obtained contact angle and surface free energy results with values presented in the literature.

I strongly recommend studying the literature; then, there will be less self-citation, especially in the Introduction section.

Leon, J. M., & Bukovac, M. J. (1978). Cuticle Development and Surface Morphology of Olive Leaves with Reference to Penetration of Foliar-applied Chemicals¹. *Journal of the American Society for Horticultural Science*, 103(4), 465-472.

We tried to follow your recommendations. I wish more people would get into determining the SFE of leaf surfaces which requires measurement of contact angles of different reference liquids, but there are only few reports and most of the were published by our group.

7. You should emphasise the separation of leaf surface structure into macrostructure and microstructure because of some inaccuracies. The author wrote that the surface roughness is low (L.130), and then they show the surface topography in Fig. 6e.

Reference to surface roughness in L130 is made for supporting the suitability of calculating its SFE from contact angle measurements. We added the prefix macro- as suggested for avoiding confusion.

Minor comments:

1. In the Abstract, could you explain what key functional properties you have in mind? Maybe it's worth mentioning which ones. (Lines 40-42)

Some properties have been mentioned as suggested.

2. Add Johnson, 1975, to the literature list and check the correctness of this sentence (Lines 79-80).

The reference has been included.

3. Move lines 122-125 and 130-133 from the Results section to the Methods section.

Lines were moved as suggested.

4. Add Jeffree, 2006, to the literature list (Lines 156, 162-163).

The reference has been included.

5. In line 198, you wrote that the yellow-red colour relates to rather hydrophobic, which sounds like you're unsure about. Am I right? Yes, you are right

6. Use space between values and units (Lines 190-191).

This was modified as suggested.

7. At the beginning of the Discussion section, you use twice the same information about UV-B protection (L. 261 and L. 267).

The sentence has been removed as suggested.

8. Add Karabourniotis et al. 1995, to the literature list (L. 268)

The reference has been included.

9. In the Method section, add information about how long the samples were kept in the fridge.

This information was added as suggested.

10. Add Khayet and Fernández 2015 to the literature list or change it to Fernández and Khayet 2015 if it is correct.

This mistake was corrected.

11. Always use a lower or upper case letter to indicate the number of samples (n = 20 L. 650; N = 30 L. 645). This was corrected as suggested.

12. The determination of surface free energy should be presented as a subsection.

This was modified as suggested.

13. Could you add the methodology of polarity calculation in the Methods section?

This has been included as suggested.

14. In Table 2S, use capital letters in line captions.

This was modified as suggested.

15. In Fig. S2, add scale bar.

This was modified as suggested.

Reviewer #2 (Remarks to the Author):

The manuscript entitled "Olive leaf wettability: from nanoscale to macroscale with focus on trichomes" by Victoria Fernández and colleagues analyzes an important property of the leaf surface by different methods. The subject is interesting, but the manuscript needs to be improved to be suitable for publication.

Dear Reviewer 2, thank you for your constructive revision of our draft which help us improve its quality.

We provide a detailed response to your comments below:

Lines 33-34: the statement is too general to be included in the Abstract; previous papers reported the same distribution of trichomes - predominantly on the lower epidermis (Koudounas et al., 2015 - cited in References in this manuscript); do the authors know any example of *Olea* sp. where the trichomes are mostly located on the upper epidermis, in mature leaves?

Thank you for your comment. We now slightly changed this opening sentence but still understood that the broad biological scope of this journal made it necessary to begin with a general statement on the relevance of hairs in biological systems.

Introduction - the first paragraph: the description of the structure of the epidermis and the cuticle is a bit simplistic, rather at the level of a university course than a scientific paper; I recommend the authors to describe in more detail the chemical composition of the cuticle because it has an essential role in the investigations carried out. The following papers can be used as references:

1. Li-Beisson, Y., Verdier, G., Xu, L., & Beisson, F. (2016). Cutin and suberin polyesters. *eLS*, 1-12.

2. Philippe, G., Sørensen, I., Jiao, C., Sun, X., Fei, Z., Domozych, D. S., & Rose, J. K. (2020). Cutin and suberin: assembly and origins of specialized lipidic cell wall scaffolds. *Current Opinion in Plant Biology*, 55, 11-20.

Thank you for your suggestions, Accordingly, the references have been included in the draft and more details are provided on cutin and wax composition. However, note this paper is not about the qualitative and quantitative analysis of olive leaf chemical constituents and this is why we did not provide more information on this in the previous version of the draft.

line 107: there is a discrepancy between the title of the chapter "Macroscopic olive leaf wettability" and the references to Figures 1A and D - obtained at SEM. These are not macroscopic observations!

We removed the term "macroscopic" from the heading for avoiding confusion.

lines 109-110: "contact angle measurements were measured" - repetition, please rephrase. This has been modified as suggested.

lines 112 and 117: how did you calculate the density of trichomes per mm²? How many fields were measured?

Images were collected from 10 leaves (with 2 different areas per leaf) as now indicated. The number of trichomes was grossly counted in the images using ImageJ software.

line 118: how did you estimate 9 ± 5 % trichome coverage? Images were collected from 10 leaves a. Trichome densities were grossly (on the lower side) counted in the images using ImageJ software.

line 144: "Olive leaf surface structure and chemical composition" - I did not find in the content references to the chemical composition of the leaf surface (the references in line 171 to the possibility of the presence of polysaccharides in the structure of the cuticle do not think that it can be considered chemical composition).

We understand your concern and hence we now included the term "chemical heterogeneity" in the heading, because both the microscopic stains utilized for OM and TEM and the immersion of leaves in cellulase and pectinase provide hints about chemical composition at least for polysaccharides (but also waxes). We hope you find this change reasonable.

line 147: "Figures 2 to 4" - figure 4 is not obtained by TEM or light microscope (it is an AFM image).

This mistake has been corrected as suggested.

line 148: "Toloudine (correct Toluidine

This mistake has been corrected as suggested.

Blue which provides cell wall polysaccharides a blue color" - on the sections made on plant material included in resins, the color obtained for polysaccharides is purple; only on manually sectioned plant material, where the sections are thicker (usually over 50-60 μm), the obtained color is blue. How thick are your sections? I recommend the following paper as a reference:

1. Ribeiro, V. C., & Leitão, C. A. E. (2020). Utilisation of Toluidine blue O pH 4.0 and histochemical inferences in plant sections obtained by free-hand. *Protoplasma*, 257(3), 993-1008.

This has been included as suggested.

line 198: "No significant correlation" - how did you come to the conclusion that the correlation is not significant?

Following your suggestion, we improved the sentences and removed the terms "no significant correlation" by explaining what we actually meant to stress from the AFM frequency images.

lines 237-239: "were immersed in solutions containing cellulose and pectin degrading enzymes (i.e., cellulase and pectinase, respectively) for 8 days and then observed potential surface changes by SEM" - don't you consider that incubation in aqueous solution for 8 days (!!) of a plant material without fixation or preservation can alone induce significant structural changes? And were the control samples incubated in the same conditions (without enzymes)? Or were fresh leaf samples used as a control? In this case, a comparison cannot be made.

We clarified this in the materials and methods section. We used as control leaves immersed in water only with Na-azide. We employed with procedure before with other plant spp. (e.g., Plant Physiol., 166(1), 168-180) and control and enzymatically treated leaf samples are comparable in experimental terms

lines 387-388: "trichomes are composed by ca. 50 μm long, elongated cells and have a large size." - this statement appears in the Discussions, but in the Results there are no results of the measurements performed on the trichomes.

This information has been now provided and the and properly addressed in the draft.

line 388: what does "macroscopic nature" mean regarding trichomes?

This has been modified with "large size"

line 595: "eventually evaluated" - please specify exactly the investigations carried out.

The word "eventually" has bene removed.

lines 607-609: please specify exactly how the SEM images in the manuscript were made; did you use low vacuum with gold coated samples?

This is described in the Materials and Methods Section.

What was the thickness of the gold layer applied?

We generally analyzed leaves under a variable pressure SEM with not Au sputtering. If there was Au sputtering this is 10 nm thick. We analyzed then with 2 different variable pressure SEMs but always fresh. These leaves are quite tough and resistant to dehydration, etc. (unlike other species we are dealing with which are quite perishable)

On lines 597 -598 it is specified "The upper (adaxial) and lower (abaxial) surface of intact, fresh (i.e., with no further preparation) olive leaves was directly analyzed by AFM and SEM." Has coating/drying been achieved or not for SEM observations?

Leaves were generally observed fresh and often with now sputtering.

line 622: what were ultra-thin sections made with? Yes, approx. 100nm thick.

With glass or diamond knives? With diamond blades

The thickness of the sections must be approximated using the color scale available for ultramicrotomes.

The ultramicrotome used is described in the Methods Section.

line 632: where do the length and diameter of the peltate trichomes appear in the Results? It is necessary to detail how many measurements were made, from how many leaves the trichomes were analyzed and to present the concrete results obtained.

This has been better described following your suggestion.

line 678: "were air-dried prior to direct SEM observation" - if these samples were air-dried (which is known to induce significant plant tissue artifacts) was the control sample dried or not? I recommend the paper below to see what changes can be induced by air drying in plant samples for SEM:

1. Bhattacharya, R., Saha, S., Kostina, O., Muravnik, L., & Mitra, A. (2020). Replacing critical point drying with a low-cost chemical drying provides comparable surface image quality of glandular trichomes from leaves of *Millingtonia hortensis* L. f. in scanning electron micrograph. *Applied Microscopy*, 50(1), 1-6.

We now quoted this interesting paper. We analysed by SEM many olive leaves of different age and gained evidence there are quite tough compared to other plant materials we dealt with before.

Reviewer #3 (Remarks to the Author):

In this work, Fernández et al. analyzed the wettability and macro- and micro-scale features of olive leaves. Combining different approaches, trichome surfaces have been found to be chemically (hydrophilic-hydrophobic) heterogeneous at the nano-scale. With it, the authors concluded that the chemical heterogeneity of trichomes may provide key functional properties to biological surfaces.

While this work has its merits and the authors provided adequate evidence for their statements, the writing can be confusing and challenging to comprehend. Despite my diligent efforts in re-reading the manuscript several times, I am left with lingering uncertainty about whether I may have inadvertently missed any critical details. The writing style hiding crucial details among expansive background information did not work in my favor.

Dear Reviewer 3,

Thank you for your comments which help us improve the quality of our draft. A detailed reply is provided below:

Since trichome is the correct botany term it should be used throughout the text instead of "hair".

We understand your concern but believe that at least using the term "hair" in the Abstract, Keywords and few places where the word "trichome" appears is helpful, provided the general biological scope of Communications Biology. Note that many Plant Science papers refer to "hairs" which may not be correct, but enhances visibility for a broader biology readership. We thank you for your understanding.

Line 56 – "aprovide"? This mistake has been corrected.

Lines 61 – 63 – Which are these "bioinspired materials"? Which "major functional roles" can the wetting properties play? It would be interesting to explore the reason for studying wettability using olive more. For me, this information is not clear throughout the manuscript.

These sentences have been improved as suggested.

Lines 86 – 87 – Which are the compounds that absorb the UV radiation?

Flavonoids and phenols. Please, read the quoted papers by Karabournioutis and co-workers who analyzed this in detail.

Line 109 – “static (with liquids of different nature)” Describe the liquids used in the experiment and why.

Please, see the improved Methods section and the references therein in which this has been described.

Line 174 – AFM must be defined in the first appearance in the text.

This has been modified as suggested.

Line 242 – “adaxial trichomes” maybe the correct term here is abaxial trichomes.

This has been modified as suggested.

Fig 2 – “LT, abaxial leaf surface.” I think that should be “abaxial leaf surface trichome,” right? YES; Although the abbreviations supposedly point to each of the structures, it would be interesting to delimit/outline areas where each of the structures shown is located. It is worth remembering that microscopy images are quite difficult for readers who are not used to them. This is described in the Fig. Caption. “lower (abaxial) epidermis; UT, adaxial leaf surface trichome; LT, abaxial leaf surface; CW”

Fig 3 - The assembly of this figure needs to be improved. The images are larger than the frames; lines overlap previous elements of the pictures, there is no standardization of the position of the labels, etc.

This has been modified as suggested.

Fig 4 – 6- I think the legends and the text associated with these pictures must be improved, considering people who have never worked with this technique.

This has been modified as suggested.

Reviewer #1:

-“Dear Authors,

I have now carefully read the manuscript entitled Olive leaf wettability: from nanoscale to macroscale, focus on trichomes. This study's main objective was characterising the olive leaf's surface as a plant material. The manuscript is well-written and contains an interesting wide range of microscopic examination results, which deserve publication in a biological journal. However, I think some comments should be considered before publication. Therefore, my decision is a revision”-.

Dear **Reviewer 1**, thank you for the revision of our manuscript and for your detailed comments that we attempted to implement in the revised version we now submitted, changes being highlighted in yellow color.

We provide a detailed reply to your concerns below:

-“1. The authors use a pseudo-equilibrium contact angle but do not explain which angle value they consider equilibrium”-.

This is a good question, especially when dealing with plant materials having trichomes (e.g., many leaves from *Quercus* spp.) which in our experience, sometimes lead to initially high water and glycerol contact angles (CA) which decrease sometimes quite significantly after few seconds. We took the CA values after depositing drops and noting that they were stable (in general, we took the values approx. 10 sec after the deposition of the water and glycerol drops). For estimating pseudo-equilibrium CAs of any liquid it is important to measure drops which are static and we always check their performance when measuring a new leaf sample. For instance, measuring glycerol CAs with the abaxial side of *Quercus ilex* leaves required several minutes to reach stable drops (see Fernández et al. 2014. *Plant Physiol.*, 166(1), 168-180).

The procedure described for measuring pseudo-equilibrium contact angles is a standard practice, but in our opinion, plant materials are quite tricky and require special attention for analysing their wettability properly. We now made a remark on this on the Materials and methods section for the sake of clarity (Line (L) 666).

-“2. To check the correctness of the calculations of the surface energy of olive leaves, it is required to know the values of both acid-base components, i.e. the electron acceptor (positive) γ^+ and the electron donor (negative) γ^- , but the authors do not provide these values. I believe these values need to be included in the revised manuscript”-.

Thank you for your comment. You are fully right regarding the calculation of the acid-base surface free energy (SFE) component, but we avoided to provide them not to come up with a larger table 2. We now provide such values as suggested.

“-3. I need clarification about the age of the leaves taken for experiments. Sometimes, the authors use the terms old and young leaves, but they need to write how they examined the age of the leaves or which leaves they consider young and which old”-.

Thank you for your comment. We refer to approx. 3 months-old leaves as young and >1 year-old as “old”/mature leaves, as described in the first paragraph of the Materials and Methods section (L618-621). We made efforts to clarify the age of leaves specially in the Materials and Methods section, but most measurements correspond to 6 months-old leaves.

-“4. In the methods section, there needs to be more information about the method for determining the surface energy of olive leaves. I don't think there's any reference to relevant literature. Khayet and Fernández 2015 are not the authors of the method, so appropriate citations should be included in the methods section. The part that should be here is in the Results section (Lines 122-125 or 130-133). Moreover, the authors did not measure the values of liquid surface tension; they only used literature values. It is better to provide the original literature (L. 649)”-.

We tried to follow your recommendation and moved some lines to the Materials and Methods section. In the previous version, we avoided to include all the references of the 3-liquids method again in the Methods Section (note that they had been quoted in the Discussion before) for not being iterative and reducing the length of the text.

On the other hand, the liquid surface tension values employed are means of all the reference data we were able to gather from the literature by Fernández and Khayet (2015) and here are hence quoting this work concerning reference liquid surface tension values used for calculations.

Also, following your suggestion a sentence was included to describe the SFE calculation method used, including the related quotations.

In summary, the whole Methods paragraph between L674-697 has been modified to include the Van Oss and co-workers' references and the rest of your suggestions.

“-5. The study's first hypothesis that the abaxial and adaxial parts of leaves have different wettability is evident, especially since such studies have already been presented in numerous kinds of literature, for example, Brewer and Nunez 2007; Holder 2007; Papierowska et al. 2018. Brewer, C. A., & Nunez, C. I. (2007). Patterns of leaf wettability along an extreme moisture gradient in western Patagonia, Argentina. *International Journal of Plant Sciences*, 168(5), 555-562.

Holder, C. D. (2007). Leaf water repellency as an adaptation to tropical montane cloud forest environments. *Biotropica*, 39(6), 767-770.

Papierowska, E., Szporak-Wasilewska, S., Szewińska, J., Szatyłowicz, J., Debaene, G., & Utratna, M. (2018). Contact angle measurements and water drop behavior on leaf surface for several deciduous shrub and tree species from a temperate zone. *Trees*, 32, 1253-1266”-.

Thank you for your comments. We included all the references suggested in the Introduction.

We now made reference to trichomes in the first hypothesis, but however, do not take anything for granted whenever we compare e.g., abaxial versus adaxial leaf surfaces of the same spp. Note that e.g., the water CA hysteresis of the upper and lower leaf side of 6-month old olive leaves is quite similar, despite there are only few peltate trichomes in the upper side which provided evidence that we should not trust our primary intuition after observing leaf surface features by e.g., scanning electron microscopy. We are missing the combined contribution of structure and chemical composition for leaf wettability and may be facing variable challenges we ignore by

only trusting on the topography of the samples. This is what we are currently trying to estimate more in detail and why we posed such apparently simple first hypothesis.

-“6. Leon and Bukovac (1978) investigated the wettability of olive leaves and presented the results of contact angle measurements. They obtained higher contact angle values in both adaxial and abaxial leaf parts (106 and 125 degrees, respectively) than presented in this manuscript. Could you explain why the results are so different? Do you think it depends on the conditions in which they grow? If so, add information about the location, climatic and meteorological conditions, etc”-.

-“It would be interesting if the discussion section included comparing the obtained contact angle and surface free energy results with values presented in the literature.

I strongly recommend studying the literature; then, there will be less self-citation, especially in the Introduction section.

Leon, J. M., & Bukovac, M. J. (1978). Cuticle Development and Surface Morphology of Olive Leaves with Reference to Penetration of Foliar-applied Chemicals¹. *Journal of the American Society for Horticultural Science*, 103(4), 465-472”-.

Dear Referee,

we were glad to read the paper by Leon and Bukovac (1978) on ‘**Manzanillo**’ olive leaves which I did not come across before. I read it back and forth and failed to clearly understand how old were the leaves for which water CAs were measured. I also failed to be clear about the volume of drops (which is a key factor) they employed and what did the mean with “advancing contact angles”. I also wonder if drops were actually determined in pseudo-equilibrium and were stable as the ones we measured.

Note that we have measured ‘**Arbequina**’ olive leaves in different seasons and from different areas of East and Southern Spain and also from Talca (Chile), and we generally got mean CA values within a similar wettability range, provided that leaves of comparable age were analysed (younger leaves are less wettable specially regarding their abaxial side).

We can hence not fully interpret the results reported by Leon and Bukovac (1978), but aware that they analysed **var. Manzanillo** and that we focussed on **var. Arbequina**. Working with 4 different corn varieties, we recently observed potentially major leaf wettability performance variations among cvs. of the same species (see Henningsen et al. 2023. *Physiol. Plantarum* 175(6), e14093).

It is more likely that the growing conditions in different world locations influence the properties of leaf surfaces, but there is no way to safely compare our leaf wettability results with the ones reported by Leon and Bukovac (1978) for the reasons explained above.

We now quoted this paper in the introduction and discussion emphasising we cannot safely compare our results with those of ‘Manzanillo’ olive leaves described by Leon and Bukovac (1978).

On the other hand, the water retention data provided by the interesting Leon and Bukovac (1978) paper which are within a similar range for the adaxial and abaxial leaf side, somehow support our water contact angle hysteresis measurements, but we are not discussing about this with more detail for the reasons described above.

In summary, we did efforts to explain our results and we are not fully able to interpret the wettability measurements performed by Leon and Bukovac (1978) at least due to variety, measuring and environmental differences between our studies (L278-285).

We tried to follow your recommendations. I wish more people would get into determining the SFE of leaf surfaces of different species which requires measurement of contact angles of various reference liquids, but there are only few reports and most of the were published by our Group.

-“7. You should emphasise the separation of leaf surface structure into macrostructure and microstructure because of some inaccuracies. The author wrote that the surface roughness is low (L.130), and then they show the surface topography in Fig. 6e”-.

Reference to surface roughness in is made for supporting the suitability of calculating its SFE from contact angle measurements. We added the prefix macro- as suggested for avoiding confusion (L121).

Minor comments:

1. In the Abstract, could you explain what key functional properties you have in mind? Maybe it's worth mentioning which ones. (Lines 40-42)

Some properties have been mentioned as suggested (L40-41).

2. Add Johnson, 1975, to the literature list and check the correctness of this sentence (Lines 79-80).

The reference has been included.

3. Move lines 122-125 and 130-133 from the Results section to the Methods section.

Lines were moved as suggested.

4. Add Jeffree, 2006, to the literature list (Lines 156, 162-163).

This reference has been included in the quotation list, as suggested .

5. In line 198, you wrote that the yellow-red colour relates to rather hydrophobic, which sounds like you're unsure about. Am I right? Yes, you are right

6. Use space between values and units (Lines 190-191).

This was modified as suggested.

7. At the beginning of the Discussion section, you use twice the same information about UV-B protection (L. 261 and L. 267).

The sentence has been removed as suggested.

8. Add Karabourniotis et al. 1995, to the literature list (L. 268)

The reference has been included.

9. In the Method section, add information about how long the samples were kept in the fridge.

This information was added as suggested.

10. Add Khayet and Fernández 2015 to the literature list or change it to Fernández and Khayet 2015 if it is correct.

This mistake was corrected.

11. Always use a lower or upper case letter to indicate the number of samples ($n = 20$ L. 650; $N = 30$ L. 645). This was corrected as suggested.

12. The determination of surface free energy should be presented as a subsection.

This was modified as suggested.

13. Could you add the methodology of polarity calculation in the Methods section?

This has been included as suggested.

14. In Table 2S, use capital letters in line captions.

This was modified as suggested.

15. In Fig. S2, add scale bar.

This was modified as suggested.

Reviewer #2:

-“The manuscript entitled "Olive leaf wettability: from nanoscale to macroscale with focus on trichomes" by Victoria Fernández and colleagues analyzes an important property of the leaf surface by different methods. The subject is interesting, but the manuscript needs to be improved to be suitable for publication”.

Dear Reviewer 2,

thank you for your constructive revision of our draft which help us improve its quality.

We provide a detailed response to your comments below and highlighted the changes in the draft in yellow color so that they are easy to identify:

-“Lines 33-34: the statement is too general to be included in the Abstract; previous papers reported the same distribution of trichomes - predominantly on the lower epidermis (Koudounas et al., 2015 - cited in References in this manuscript); do the authors know any example of *Olea* sp. where the trichomes are mostly located on the upper epidermis, in mature leaves?”

Thank you for your comment. We now slightly changed this opening sentence but still understood that the broad biological scope of this journal made it necessary to begin with a general statement on the relevance of hairs in biological systems.

-“Introduction - the first paragraph: the description of the structure of the epidermis and the cuticle is a bit simplistic, rather at the level of a university course than a scientific paper; I recommend the authors to describe in more detail the chemical composition of the cuticle because it has an essential role in the investigations carried out. The following papers can be used as references:

1. Li-Beisson, Y., Verdier, G., Xu, L., & Beisson, F. (2016). Cutin and suberin polyesters. *eLS*, 1-12.
2. Philippe, G., Sørensen, I., Jiao, C., Sun, X., Fei, Z., Domozych, D. S., & Rose, J. K. (2020). Cutin and suberin: assembly and origins of specialized lipidic cell wall scaffolds. *Current Opinion in Plant Biology*, 55, 11-20”-.

Thank you for your suggestions, Accordingly, the references have been included in the draft and more details are provided on cutin and wax composition (Line (L)53-57). However, note this paper is not about the qualitative and quantitative analysis of olive leaf chemical constituents and this is why we did not provide more information on this in the previous version of the draft.

-“line 107: there is a discrepancy between the title of the chapter "Macroscopic olive leaf wettability" and the references to Figures 1A and D - obtained at SEM. These are not macroscopic observations!”

We removed the term “macroscopic” from the heading for avoiding confusion.

-“lines 109-110: "contact angle measurements were measured" - repetition, please rephrase”.

This has been modified as suggested.

-“lines 112 and 117: how did you calculate the density of trichomes per mm²? How many fields were measured?”-

Images were collected from 10 leaves (with 2 different areas per leaf) as now indicated. The number of trichomes was grossly counted in the images using ImageJ software.

-“line 118: how did you estimate 9 ± 5 % trichome coverage?”

Images were collected from 10 leaves a. Trichome densities were grossly (on the lower side) counted in the images using ImageJ software.

-“line 144: "Olive leaf surface structure and chemical composition" - I did not find in the content references to the chemical composition of the leaf surface (the references in line 171 to the possibility of the presence of polysaccharides in the structure of the cuticle do not think that it can be considered chemical composition)”-.

We understand your concern and hence we now included the term “chemical heterogeneity” in the heading, because both the microscopic stains utilized for OM and TEM and the immersion of leaves in cellulase and pectinase provide hints about chemical composition at least for polysaccharides (but also waxes). We hope you find this change reasonable.

-“line 147: "Figures 2 to 4" - figure 4 is not obtained by TEM or light microscope (it is an AFM image)”-.

This mistake has been corrected as suggested.

-“line 148: "Toloudine (correct Toluidine)”-

This mistake has been corrected as suggested.

-“ Blue which provides cell wall polysaccharides a blue color" - on the sections made on plant material included in resins, the color obtained for polysaccharides is purple; only on manually sectioned plant material, where the sections are thicker (usually over 50-60 μm), the obtained color is blue. How thick are your sections? I recommend the following paper as a reference: 1. Ribeiro, V. C., & Leitão, C. A. E. (2020). Utilisation of Toluidine blue O pH 4.0 and histochemical inferences in plant sections obtained by free-hand. *Protoplasma*, 257(3), 993-1008”-.

This interesting reference has been included as suggested.

-“line 198: "No significant correlation" - how did you come to the conclusion that the correlation is not significant?”-

Following your suggestion, we improved the sentences and removed the terms “no significant correlation” by explaining what we actually meant to stress from the AFM frequency images (L186-193).

-“lines 237-239: "were immersed in solutions containing cellulose and pectin degrading enzymes (i.e., cellulase and pectinase, respectively) for 8 days and then observed potential surface changes by SEM” - don't you consider that incubation in aqueous solution for 8 days (!) of a plant material without fixation or preservation can alone induce significant structural changes? And were the control samples incubated in the same conditions (without enzymes)? Or were fresh leaf samples used as a control? In this case, a comparison cannot be made”-.

We better clarified this in the Methods (L710-718) and also in the Results (L227) section. We used as control leaves immersed in water only with Na-azide. We employed with procedure before with other plant spp. (e.g., *Plant Physiol.*, 166(1), 168-180) and control and enzymatically treated leaf samples are comparable in experimental terms

-“lines 387-388: "trichomes are composed by ca. 50 μm long, elongated cells and have a large size." - this statement appears in the Discussions, but in the Results there are no results of the measurements performed on the trichomes”-.

This information has been now provided and the and properly addressed in the draft.

-“line 388: what does "macroscopic nature" mean regarding trichomes?”-

This has been modified with “large size”

--line 595: "eventually evaluated" - please specify exactly the investigations carried out”-.

The word “eventually” has been removed.

-“lines 607-609: please specify exactly how the SEM images in the manuscript were made; did you use low vacuum with gold coated samples?”-

This is described in the Materials and Methods Section.

-“What was the thickness of the gold layer applied?”-

We generally analyzed leaves under a variable pressure SEM with not Au sputtering. If there was Au sputtering this is 10 nm thick. We analyzed then with 2 different variable pressure SEMs but always fresh. These leaves are quite tough and resistant to dehydration, etc. (unlike other species we are dealing with which are quite perishable)

-“On lines 597 -598 it is specified "The upper (adaxial) and lower (abaxial) surface of intact, fresh (i.e., with no further preparation) olive leaves was directly analyzed by AFM and SEM." Has coating/drying been achieved or not for SEM observations?”-

Leaves were generally observed fresh (with all SEMs used) and often with now sputtering by FESEM).

-“line 622: what were ultra-thin sections made with?”

Yes, approximately 100 nm thick.

-“With glass or diamond knives?”-

Yes, with diamond blades.

-“The thickness of the sections must be approximated using the color scale available for ultramicrotomes”.

The ultramicrotome used is described in the Methods Section.

-“line 632: where do the length and diameter of the peltate trichomes appear in the Results? It is necessary to detail how many measurements were made, from how many leaves the trichomes were analyzed and to present the concrete results obtained”-.

This has been better described following your suggestion.

-“line 678: "were air-dried prior to direct SEM observation" - if these samples were air-dried (which is known to induce significant plant tissue artifacts) was the control sample dried or not? I recommend the paper below to see what changes can be induced by air drying in plant samples for SEM:

1. Bhattacharya, R., Saha, S., Kostina, O., Muravnik, L., & Mitra, A. (2020). Replacing critical point drying with a low-cost chemical drying provides comparable surface image quality of glandular trichomes from leaves of *Millingtonia hortensis* L. f. in scanning electron micrograph. *Applied Microscopy*, 50(1), 1-6”-.

We now quoted this interesting paper. We analysed by SEM many olive leaves of different age and gained evidence there are quite tough compared to other plant materials we dealt with before.

Reviewer #3:

-“In this work, Fernández et al. analyzed the wettability and macro- and micro-scale features of olive leaves. Combining different approaches, trichome surfaces have been found to be chemically (hydrophilic-hydrophobic) heterogeneous at the nano-scale. With it, the authors concluded that the chemical heterogeneity of trichomes may provide key functional properties to biological surfaces.

While this work has its merits and the authors provided adequate evidence for their statements, the writing can be confusing and challenging to comprehend. Despite my diligent efforts in re-reading the manuscript several times, I am left with lingering uncertainty about whether I may have inadvertently missed any critical details. The writing style hiding crucial details among expansive background information did not work in my favour”.

Dear Reviewer 3,

Thank you for your comments which help us improve the quality of our draft. Changes in the manuscript are now highlighted in yellow color so that they can be easily spotted.

A detailed reply is provided below:

-“Since trichome is the correct botany term it should be used throughout the text instead of “hair”-.

We understand your concern but believe that at least using the term “hair” in the Abstract, Keywords and few places where the word “trichome” appears is helpful, provided the general biological scope of Communications Biology. Note that many Plant Science papers refer to “hairs” which may not be correct, but enhances visibility for a broader biology readership. We thank you for your understanding.

-“Line 56 – “aprovide”?”-

This mistake has been corrected.

-“Lines 61 – 63 – Which are these “bioinspired materials”? Which “major functional roles” can the wetting properties play? It would be interesting to explore the reason for studying wettability using olive more. For me, this information is not clear throughout the manuscript.

These sentences have been improved as suggested.

We now provided the reason to study olive leaves in L247-251, as suggested

-“Lines 86 – 87 – Which are the compounds that absorb the UV radiation?”-

Flavonoids and phenols. Please, read the quoted papers by Karabournioutis and co-workers who analyzed this in detail.

-“Line 109 – “static (with liquids of different nature)” Describe the liquids used in the experiment and why”.

-We improved improved Methods section and the references therein as suggested

-“Line 174 – AFM must be defined in the first appearance in the text”-.

This has been modified as suggested.

-“Line 242 – “adaxial trichomes” maybe the correct term here is abaxial trichomes”-.

This has been modified as suggested.

-“Fig 2 – “LT, abaxial leaf surface.” I think that should be “abaxial leaf surface trichome,” right?”-

Yes, you are right

-“Although the abbreviations supposedly point to each of the structures, it would be interesting to delimit/outline areas where each of the structures shown is located. It is worth remembering that microscopy images are quite difficult for readers who are not used to them”.

This is described in the Fig. Caption. “lower (abaxial) epidermis; UT, adaxial leaf surface trichome; LT, abaxial leaf surface; CW”

-“Fig 3 - The assembly of this figure needs to be improved. The images are larger than the frames; lines overlap previous elements of the pictures, there is no standardization of the position of the labels, etc.”-

This has been modified as suggested.

-“Fig 4 – 6- I think the legends and the text associated with these pictures must be improved, considering people who have never worked with this technique”.

This has been modified as suggested.

REVIEWERS' COMMENTS:

Reviewer #1 (Remarks to the Author):

Dear Authors,
I appreciate your work to improve the manuscript and to answer all my comments.

Reviewer

Reviewer #2 (Remarks to the Author):

I checked the revised manuscript, as well as the answers given by the authors to the 3 reviewers. From my point of view, most of the requested changes have been made, so the manuscript can be published.

Reviewer #3 (Remarks to the Author):

I have read the current version of this manuscript and find it was improved, as compared to the original version. All the major points raised in the previous round of revision were addressed by the authors by modifying the text or adding figures and tables. It seems to me that the manuscript is well structured, as presented in its current version.